# Enhancing LLM Planning for Robotics Manipulation through Hierarchical Procedural Knowledge Graphs

**Jiacong Zhou**[1*]   **Jiaxu Miao**[2*]   **Xianyun Wang**[2,3]   **Jun Yu**[2,3†]

[1]School of Computer Science, Hangzhou Dianzi University
[2]The School of Intelligence Science and Engineering, Harbin Institute of Technology, Shenzhen
[3]Pengcheng Laboratory
jczhou@hdu.edu.cn  yujun@hit.edu.cn

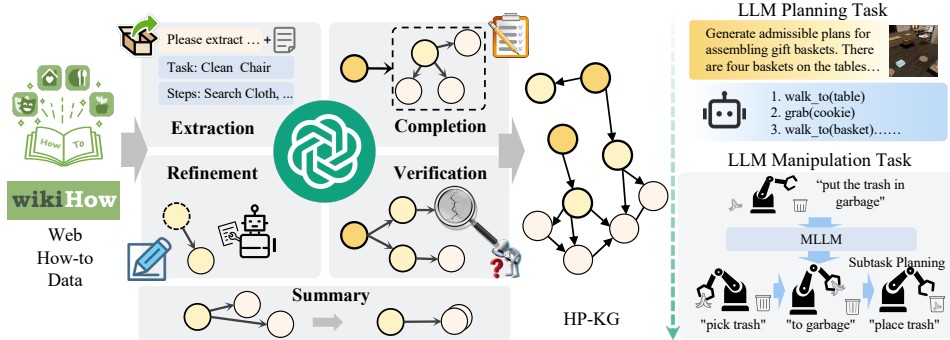

Figure 1: Overview of our work. We propose a large-scale Procedural Knowledge Graph constructed from web how-to data, which can significantly enhance the planning capabilities of LLMs in robot manipulation tasks.

## Abstract

Large Language Models (LLMs) have shown the promising planning capabilities for robotic manipulation, which advances the development of embodied intelligence significantly. However, existing LLM-driven robotic manipulation approaches excel at simple pick-and-place tasks but are insufficient for complex manipulation tasks due to inaccurate procedural knowledge. Besides, for embodied intelligence, equipping a large scale LLM is energy-consuming and inefficient, which affects its real-world application. To address the above problems, we propose Hierarchical Procedural Knowledge Graphs (**HP-KG**) to enhance LLMs for complex robotic planning while significantly reducing the demand for LLM scale in robotic manipulation. Considering that the complex real-world tasks require multiple steps, and each step is composed of robotic-understandable atomic actions, we design a hierarchical knowledge graph structure to model the relationships between tasks, steps, and actions. This design bridges the gap between human instructions and robotic manipulation actions. To construct HP-KG, we develop an automatic knowledge graph construction framework powered by LLM-based multi-agents, which eliminates costly manual efforts while maintaining high-quality graph structures. The resulting HP-KG encompasses over 40k activity steps across more than 6k household tasks, spanning diverse everyday scenarios. Extensive experiments demonstrate that small scale LLMs (7B) enhanced by our HP-KG significantly improve the planning capabilities, which are stronger than 72B LLMs only. En-

---

[*]Equal contribution.
[†]Corresponding author.

39th Conference on Neural Information Processing Systems (NeurIPS 2025).

couragingly, our approach remains effective on the most powerful GPT-4o model. Our code and data will be publicly available[3].

# 1   Introduction

Embodied AI [1, 2] refers to AI that is integrated into physical systems, such as robots, enabling them to interact with the physical world with the abilities of perception, reasoning, planning, and execution. Recent achievements of Large Language Models (LLMs) [3–8] have shown the remarkable capabilities for robotic task planning [9, 10], greatly advancing the development of embodied AI. These models usually serve as high-level planners to decompose human instructions into executable sub-goals [11, 9, 12, 10, 13], while relying on pre-defined skills [14–16] for execution.

Recent studies [17, 18] show that such LLM-driven planning methods usually generate unrealistic or logically inconsistent planning steps due to lacking procedural commonsense, especially in complex manipulation tasks. For instance, LLMs may ignore the agent's current physical state, thus failing to include necessary prerequisite actions (e.g., a standup action before any movement) [19], or overlook physical constraints such as the need to open a closed container before fetching items from inside [20]. Moreover, robots are often limited by a finite energy supply, while LLM-driven planners typically require large-scale models (e.g., PaLM-E [21] with 562B parameters) to possess sufficient planning capabilities in complex, long-horizon scenarios. For embodied intelligence, equipping a large scale LLM is energy-consuming and inefficient, which affects its real-world application.

To address the above problems, we propose to construct an effective Procedural Knowledge Graph to enhance LLM-driven planners. Procedural knowledge refers to the understanding of how to perform specific tasks, typically expressed as sequences of steps required to achieve a specific goal [22, 19]. Since the Procedural Knowledge Graph provides the commonsense needed for planning, injecting the correct Procedural Knowledge into LLMs can effectively enhance their reasoning accuracy. Therefore, a small-scale LLM equipped with the Procedural Knowledge Graph has sufficient capability to perform planning, alleviating the computational cost demands of embodied AI.

However, developing an effective representation of procedural knowledge to enhance robotic planning capabilities remains a challenge. Existing methods usually adopt coarse-grained task decomposition and simply represent procedural knowledge as reference documents [23, 24, 19], without specific structure design. The main challenge in robot planning lies in translating high-level goals into feasible steps, due to the domain gap between language comprehension and robotic execution. A natural observation is that procedures of robot manipulation can be divided into a finite set of atomic actions that the robot can understand and execute. These atomic actions can be combined into a series of steps, which can further be combined into tasks that are highly abstract as human instruction in the real world. Motivated by this, we design a novel and effective Hierarchical Procedural Knowledge Graph (**HP-KG**) to organize procedures into three distinct layers: tasks, steps, and actions, as shown in Figure 2. Each procedure is further enriched with textual attributes (description, name, and tips).

Furthermore, to eliminate the need for manual knowledge engineering and reduce human effort, we introduce a framework to automatically construct the procedural knowledge graph through multi-agents calibration [25, 5, 26]. Specifically, we focus on household activities in this work as they are frequently performed by everyone and represent a promising area for robotic assistance to facilitate daily living [27]. We systematically filter household-related tasks from the WikiHow corpus [28] and combine them with the BEHAVIOR dataset [27] as knowledge source. We then prompt an LLM to extract steps from each household task and complete them by generating corresponding actions and their textual attributes. Subsequently, two LLM agents are employed to iteratively verify and refine the generated procedures based on designed rules. To enrich information while reducing redundancy, we perform semantic similarity clustering and LLM-based knowledge merging.

Finally, we propose a retrieval approach to leverage the constructed HP-KG. Given a language instruction, a refined query is generated to retrieve relevant knowledge nodes based on semantic similarity. Through K-hop breadth-first search and re-ranking, it identifies the most pertinent nodes and converts their sub-graphs into textual descriptions for contextual planning. Extensive experiments on ActPlan-1K [18] and RLBench [29] demonstrate that our HP-KG enables smaller models (7B) to achieve stronger capabilities than larger models (72B) only. Encouragingly, our approach remains effective on the most powerful GPT-4o model. Overall, the contributions of our work are as follows:

---

[3]https://anonymous.4open.science/r/HP-KG-68EE/

• We design a novel hierarchical procedural knowledge graph structure that effectively formalizes complex household tasks through a structure of tasks, steps, and actions.

• We introduce a novel automated framework that leverages LLM-based multi-agents to construct hierarchical procedural knowledge graphs, eliminating the need for manual knowledge engineering.

• We present HP-KG, a large-scale procedural knowledge graph, containing 42,000+ activity steps across 6,000+ household tasks in daily scenarios.

• Our method significantly improves the planning capability (+17.64% on 7B LLMs) and reduces the scale demand for LLMs.

## 2 Related Works

**Knowledge Graph Construction.** Traditional Knowledge Graph Construction methods typically involve multiple tasks, including entity extraction [30, 31] and relation classification [32, 33], which incur substantial human effort and cost [34]. With the advancement of pre-trained language models like BERT [35], GPT-3 [36], end-to-end triplet extraction emerges as a promising paradigm [34, 37]. Contemporary graph construction approaches [38–40] leverage large language models (LLMs) [3, 5, 8] for entity extraction via prompting or fine-tuning.

**LLMs with Knowledge Graph.** Knowledge graphs (KGs), which provide structured factual representations, have emerged as a powerful tool to enhance LLM performance [41–43]. Recent advances in Knowledge Graph Augmented Generation [44–46] demonstrate that KGs can serve as external knowledge bases to provide accurate factual information, effectively enhancing the factual correctness in LLM-generated responses [47–50]. Furthermore, several studies have shown that leveraging the structural relationships inherent in KGs can enhance the reasoning capabilities of LLMs [51–53].

**Foundation Models for Robotics Manipulation.** Recent achievements of vision-language foundation models [54–56] have significantly influenced the field of robotics manipulation. These models demonstrate the potential for controlling robots to perform complex tasks. Recent studies can be broadly categorized into two paradigms: One stream adopts vision-language-action models [57, 58] such as RT-2 [59], RT-X [60], and OpenVLA [61] to directly map visual inputs and language instructions to robotic actions. The other paradigm leverages vision-language models (VLMs) to decompose high-level instructions into sub-goals [11, 9, 62–64], which are then solved through pre-defined skills [14, 65, 15, 66]. For example, Yang et al. [67] leverages Vision Language Models to generate subgoals, then employs dynamics-based planning to solve the subgoal. Our method is orthogonal to these approaches. Benefiting from the in-context understanding capabilities of vision language models, our HP-KG can integrate seamlessly with various methods' subgoal generation processes.

## 3 Approach

In this section, we first present our structure design of the Hierarchical Procedural Knowledge Graph (HP-KG) for household activities. Second, we propose an automated procedural knowledge graph construction framework for building our HP-KG. Third, we introduce a retrieval method that retrieves relevant procedural knowledge as contextual input to enhance LLM-based planning.

### 3.1 Hierarchical Structure Design for Procedural Knowledge Graph

One main challenge of robotics planning tasks is how to bridge high-level human instructions and atomic actions that are understandable by robots. Representing and organizing procedural knowledge is central to its ability to assist robotics planning effectively. A well-structured procedural knowledge graph should be easily applicable to diverse downstream applications and guide reliable execution by both human operators and robotic systems.

To this end, our procedure structure design should satisfy the following principles. First, the nodes of this knowledge graph should include comprehensive tasks and atomic actions, as well as how to effectively connect them. Second, the knowledge graph should include safety-critical guidelines to prevent potential accidents, which is essential for ensuring correct procedure execution [68]. We observe that complex procedures in household activities are composed of a set of atomic procedures (e.g., open container). Some atomic actions combine to form a simple step. A series of steps, in turn,

combine to create complex tasks that approach high-level human instructions (e.g., clean the room), naturally forming a hierarchical structure of procedural knowledge.

Based on these considerations, as Figure 2 shows, we design a simple yet effective Hierarchical Structure that encompasses multiple characteristics. To represent the hierarchical nature of procedural knowledge, we categorize procedures into three levels: tasks, steps, and actions, which correspond to complex composite procedures, simple composite procedures, and atomic procedures, respectively. A task node links to step nodes via *HasStep* edges, and step nodes link to action nodes through *HasAction* edges. The temporal sequence is captured by *NextStep* edges between steps and *NextAction* edges between actions. Additionally, each procedure contains two key attributes: *HasDescription* specifies what needs to be performed, *HasTips* provides tips and warnings.

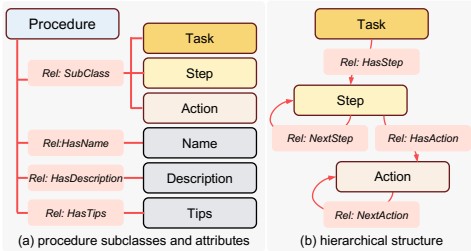

Figure 2: Overview of procedural graph structure. **(a)** Procedure subclass and their attribute relationships. **(b)** The hierarchical structure between different procedures.

## 3.2 Automatic Graph Construction Framework

To construct our HP-KG, we propose an LLM-based multi-agent framework for automatic Procedural Knowledge Graph construction. As Figure 3 shows, the framework comprises four key stages: (1) data source cleaning and processing, (2) procedures generation and completion, (3) rules-guided iterative verification and refinement, and (4) hierarchical clustering and summarization.

**Data Source Cleaning and Processing.** In our work, we utilize two main data sources. The first is the WikiHow corpus [28], one of the largest online databases containing comprehensive how-to articles across multiple domains. Each article in WikiHow typically consists of a title, a main goal, and several methods to achieve this goal, where each method contains multiple steps with detailed descriptions. The second is the BEHAVIOR-1K dataset [27], which comprises approximately 1,000 most common household activities. Each activity is defined through a BDDL (Behavior Domain Definition Language) file that specifies the activity goals and available objects in the environment.

The WikiHow corpus contains numerous articles across various domains, while only household activities are needed in this paper. Moreover, documents in the corpus may contain semantically similar activities, which could lead to redundancy in the extracted procedures. Therefore, we first filter documents by categories, retaining only household-related categories. Subsequently, we employ Deepseek-V3 [4] to determine whether each document describes a household activity based on its title and descriptions (detailed prompt template is provided in Appendix E). Finally, we cluster documents based on semantic similarity. We obtain approximately 5.8K unique household activities from the WikiHow corpus. Furthermore, we standardize input formats for BDDL-defined activities in the BEHAVIOR-1K dataset by using LLMs to generate WikiHow-style content.

**Procedures Generation and Completion.** Based on collected documents, we prompt the LLM agent as the **Procedure Generator** to extract and complete procedures. During the extraction phase, the generator extracts each activity's tasks, steps, and corresponding textual attributes (e.g., task content, task tips) from each document. According to our hierarchical graph structure, we build the interconnection between tasks and steps through the *hasStep* relation. After extraction, we design an action completion prompt to generate atomic action procedures with minimal redundancy. Using this prompt, the generator produces corresponding atomic actions and their attributes for each step procedure. These actions along with their attributes constitute action procedures, which are then linked to their respective step procedures through the *hasAction* relation. These hierarchical procedures interconnect through defined relations, forming a preliminary procedures graph.

**Rules-Guided Iterative Verification and Refinement.** Due to the inherent limitations in Large Language Models (LLMs), the generated preliminary procedures inadequately adhere to procedural principles (e.g., they contain redundant information and tend to be overly specific). To address these challenges, we propose an Iterative Verification and Refinement method to improve the quality of procedures. A **Rules-Guided Verifier** invokes multiple rule-checking models to assess the given procedure's compliance with established rules, subsequently aggregating these results to generate a

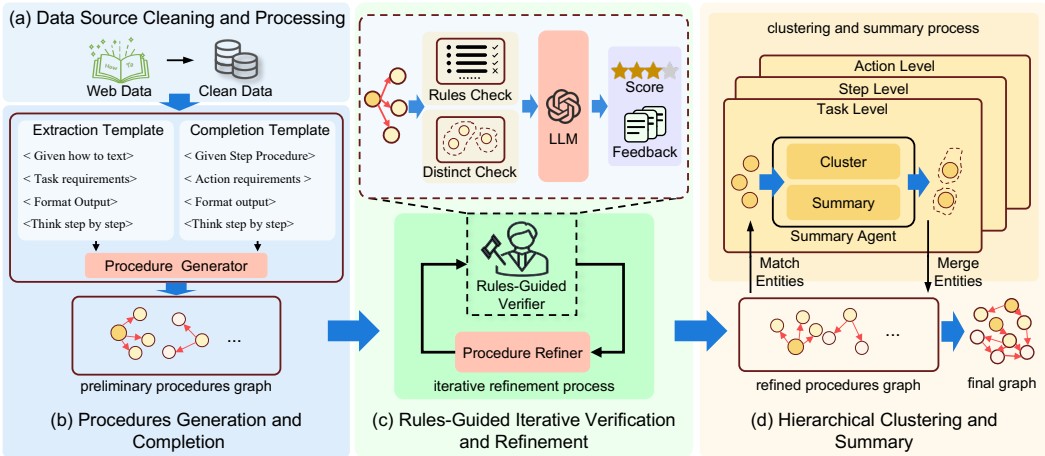

Figure 3: Overview of our automatic construction pipeline: (a) Data source cleaning and processing. (b) A **Generator** extracts steps with attributes from the data source and performs actions completion. (c) A **Verifier** and A **Refiner** iteratively validate procedures against designed rules while progressively refining them. (d) A **Summary Agent** hierarchically clusters and merges redundant procedures.

validation score and feedback. Additionally, we prompt an LLM as a **Procedure Refiner** to iteratively modify the provided procedures until the score surpasses the predefined threshold.

Specifically, based on fundamental principles of our Hierarchical Structure Design, we construct a comprehensive set of validation rules (e.g., check if content accurately describes a procedure, check if action is atomic, etc.), denoted as $\mathcal{R} = \{r_1, r_2, ..., r_n\}$, where $n$ is the total number of rules. Given a generated procedure $p_t$ and its corresponding sub-procedures $\mathbf{p_c} = (p_c^1, ..., p_c^k)$, we concatenate them into $\mathbf{p} = [p_t; p_c^1; ...; p_c^k]$ to serve as input to the Verifier, where sub-procedures represent nodes that are linked by $p_t$ via *hasStep* or *hasAction* relations. The agent then invokes a rule-checking language model $M$ to evaluate whether generated procedures $\mathbf{p}$ comply with rules in $\mathcal{R}$. For each rule $r_i \in \mathcal{R}$, the model outputs a binary decision $M(\mathbf{p}, r_i) \in \{0, 1\}$, where 1 indicates compliance and 0 indicates violation of the respective rule. Therefore, we obtain rule compliance scores, denoted as:

$$\mathbf{s}(\mathbf{p}, \mathcal{R}; M) = \Big( M(\mathbf{p}, r_i), M(\mathbf{p}, r_2), ..., M(\mathbf{p}, r_n) \Big). \tag{1}$$

To identify redundant elements across sub-procedures $\mathbf{p_c}$, the agent also employs an embedding model[4] to encode both the name and description of each procedure. It then groups sub-procedures whose pairwise embedding cosine similarity exceeds a threshold, resulting in a list of clusters $\mathbf{C} = (C_1, C_2, ...C_t)$, where $t$ is the number of groups. Furthermore, the Verifier converts rule validation results $\mathbf{s}$ and aggregation results $C$ into textual descriptions separately. It then uses these results and procedures $\mathbf{p}$ to prompt a powerful LLM (e.g., GPT-4o) to generate a score and feedback.

After the verification, we use a Procedure Refiner to modify given procedures $\mathbf{p}$. The Refiner utilizes feedbacks, textual result descriptions, and given procedures $\mathbf{p}$ as inputs to generate modified procedures $\mathbf{p}'$. These refined procedures are then iteratively verified until their scores surpass a predefined threshold. We provide the complete prompts and detailed framework rules in Appendix E.

**Hierarchical Cluster and Summary.** Although our iterative verification and refinement process enhances the quality of hierarchical procedures (e.g., tasks and their sub-steps, steps and their sub-actions), redundancy persists among procedures at the same level. To address this problem, we cluster and merge procedures of the same level via a **Summary Agent**.

We first match all procedures at a given level (e.g., task procedures) as the input of the Summary Agent. This agent uses the same embedding model to encode each procedure's name and content and computes the cos similarity between each pair of procedures. Procedures with similar semantics will be aggregated into a group $U_i$, together forming the group list $\mathbf{U} = (U_1, U_2, ...)$.

For each group $U_i$ that contains multiple procedures, the Summary Agent employs an LLM to generate comprehensive summaries based on designed prompt "Given multiple semantically similar procedures,

---

[4]https://huggingface.co/NovaSearch/stella_en_400M_v5

analyze and synthesize them into a single and actionable summary...". After the summarization process, we merge these newly generated procedures into the existing graph. We iteratively repeat the summarization process from task level to action level, ultimately deriving the final procedural graph.

### 3.3 Graph Retrieval-Augmented Planning

In this section, we propose a graph retrieval-augmented reasoning approach that utilizes HP-KG to enhance LLM planning and robotic manipulation. We aim to retrieve the top-$K$ relevant nodes at specified target levels and transform their sub-graphs into textual descriptions to facilitate contextual planning. Our retrieval-augmented method consists of (1) Node Indexing, (2) Query Retrieval, (3) Entity Expansion and Re-Ranking, (4) Procedural Graph Augmented Planning.

**Node Indexing.** For each node $n$ with textual attributes $x_n$, we apply a pre-trained embedding encoder (e.g., Zhang et al. [69]) to obtain its representation $z_n \in \mathbb{R}^d$:

$$z_n = \text{Encoder}(x_n). \tag{2}$$

**Query Retrieval.** Let $\mathcal{G} = (\mathcal{V}, \mathcal{E}, X)$ denote our HP-KG, where $\mathcal{V}$ and $\mathcal{E}$ denote the sets of nodes and edges respectively. Additionally, $X \in \mathbb{R}^{|\mathcal{V}| \times d}$ represents the node feature matrix, where $|\mathcal{V}|$ denotes the number of nodes and $d$ is the feature dimension.

We begin with a general instruction and a retrieval target level $L_{\text{target}}$. First, we prompt LLMs to extract the task objectives as query $x_q$ and apply same embdding model to encode $x_q$ as follows:

$$z_q = \text{Encoder}(x_q) \in \mathbb{R}^d. \tag{3}$$

We then retrieve the top-$k_1$ most semantically similar procedure nodes $V_{k_1}$.

$$V_{k_1} = \text{TopK}_{n \in \mathcal{V}} \text{sim}(z_q, z_n), \tag{4}$$

where $z_n$ denotes the embeddings of node $n$ and $\text{sim}(z_q, z_n)$ denotes the cosine similarity between $z_q$ and $z_n$. The TopK returns the top-$k$ nodes with highest similarity.

**Entity Expansion and Re-Ranking.** Since neighboring nodes often encapsulate complementary or contextually relevant semantic information for the initially retrieved nodes, we aim to expand our retrieved nodes to discover additional relevant procedures. Specifically, we expand the retrieved nodes $V_{k_1}$ by performing a K-hop breadth-first traversal from each retrieved node as follows:

$$V'_{k_1} = \bigcup_{n \in V_{k_1}} \{n' \mid \text{dist}(n', n) \leq K, n' \in \mathcal{V}\}, \tag{5}$$

where $\text{dist}(\cdot)$ denotes the path distance between nodes in HP-KG. Furthermore, we filter out nodes that do not match the target level $L_{\text{target}}$ from $V'_{k_1}$, obtaining a refined set:

$$V_{\text{target}} = \{n \in V'_{k_1} \mid \text{Level}(n) = L_{\text{target}}\}, \tag{6}$$

Where $\text{Level}(\cdot)$ denotes the level of node. Then, we re-rank the candidate nodes $V_{\text{target}}$ based on query $x_q$, and select the top-$k_2$ nodes $V_{k_2}$ as follows:

$$V_{k_2} = \text{TopK}_{n \in V_{\text{target}}} \text{sim}(z_q, z_n). \tag{7}$$

**Procedural Graph Augmented Planning.** After obtaining the retrieved nodes $V_{k_2}$, we aim to convert these nodes into textual context for LLM planning. To this end, we require not only the retrieved nodes but also the associated sub-procedures for each procedure node. These sub-procedures detail the sequential steps of each task and the specific actions required within each step.

Therefore, we extract associated lower-level nodes based on predefined relations (e.g., HasStep, HasAction, etc.), constructing hierarchical subgraphs where each retrieved node serves as the root. Furthermore, we convert each hierarchical subgraph into a textual description (detailed templates are in Appendix E). To enable the procedural graph augmented planning, we feed the converted textual descriptions into the LLM as additional context. The detailed algorithm is presented in Appendix B.

## 4  Experiments

In this section, we conduct comprehensive experiments on LLM planning and robotic manipulation benchmarks to evaluate the effectiveness and efficiency of our HP-KG. Extensive ablation studies are also conducted to show the effectiveness of our hierarchical structure and the construction framework.

Table 1: Comparison of task success rates (%) for zero-shot robotic manipulation on RLBench. Our proposed HP-KG improves baseline success rates when retrieving the Top-K (K=3,5) most relevant procedures (numbers in parentheses indicating absolute gains).

| Method | HP-KG | Top-K | Open_wine | Take_scale | Take_umbrella | Slide_block | Play_jenga | Average |
|---|---|---|---|---|---|---|---|---|
| | - | - | 10% | 15% | 60% | 50% | 0% | 27% |
| Voxposer [9] | ✓ | Top-3 | 40% | 25% | 60% | 45% | 0% | 34%(**+7%**) |
| | ✓ | Top-5 | 40% | 20% | 65% | 60% | 0% | 37%(**+10%**) |
| | - | - | 20% | 0% | 35% | 10% | 20% | 17% |
| MA [11] | ✓ | Top-3 | 20% | 0% | 50% | 30% | 40% | 28%(**+11%**) |
| | ✓ | Top-5 | 10% | 0% | 40% | 30% | 35% | 23%(**+6%**) |

Table 2: We compare the effectiveness of our HP-KG against Chain of Thought (COT) prompting for action planning. In both cases, we employ RVT as the action executor.

| Method | Top-K | Open_wine | Take_lid | Take_scale | Take_umbrella | Slide_block | Average |
|---|---|---|---|---|---|---|---|
| RVT [57] | - | 65% | 5% | 0% | 30% | 15% | 23% |
| RVT w/ COT | - | 70% | 10% | 15% | 35% | 5% | 27%(+4%) |
| RVT w/ HP-KG | Top-3 | 75% | 10% | 10% | 55% | 20% | 34%(**+11%**) |
| RVT w/ HP-KG | Top-5 | 85% | 5% | 15% | 50% | 10% | 33%(**+10%**) |

## 4.1 Experimental Settings

**Benchmarks.** We mainly evaluate our HP-KG on **RLBench** [29] and **Blocks Arrange** [70] tasks for robotics manipulation. Furthermore, we also evaluate our approach on **ActPlan-1K** [18] for LLM planning. Detailed environment setup is provided in the Appendix A.

- **RLBench.** This benchmark comprises 100 distinct hand-crafted manipulation tasks with varying complexity, ranging from basic target reaching and door opening to complex multi-stage operations. Each task is specified by a natural language instruction that defines the manipulation goal. In our experiments, we select a set of 6 complex tasks, each requiring multiple actions.
- **Blocks Arrange.** This task requires robots to arrange randomly located colored boxes according to given textual instructions. In our experiments, we selected 5 complex tasks with varying difficulty levels, ranging from relatively simple one like put_blocks_in_corners to more challenging ones such as stack_blocks_into_three_towers.
- **ActPlan-1K.** This dataset contains multi-modal household activity planning instances. The task in ActPlan-1K requires a MLLM to generate procedural plans $P$ given a task description $T$ and a sequence of environment images $\{I_1, I_2, ...\}$ as input. Each instance is annotated with two reference procedural plans $P_1^*, P_2^*$, where each plan consists of multi-step action sequences.

**Baselines.** In RLBench, we compare different zero-shot manipulation approaches that leverage large language models (e.g., VoxPoser [9], MA [11]). We enhance their sub-task planning process by incorporating procedural knowledge and compare the results against their baseline performance. In the ActPlan-1K benchmark, we evaluate various multi-modal large language models with different scales (e.g., GPT-4o [54], Gemini [55], Qwen2-VL [72]) both with and without HP-KG augmentation. We also leverage LLMs to generate subgoals and employ a pretrained policy [57] for execution on RLBench. In this context, we compare our HP-KG against Chain of Thought (COT)[73] method. Additionally, we compare with SayCan[71] on Blocks Arrange task.

**Metrics.** We employ Success Rate (SR) for RLBench and Blocks Arrange, while using the Longest Common Subsequence (LCS) metric for the ActPlan-1K benchmark. Detailed metrics are as follows:

- **Success Rate.** Similar to VoxPoser, we conduct 20 trials per task and calculate the success rate.
- **LCS.** On the ActPlan-1K benchmark, we use the longest common subsequence (LCS) metric to evaluate the generated plans against annotated reference plans (using the first one as default). Following [18], we encode each step with a language embedding model[5] and consider those with similarity above 0.8 as matching steps in the LCS calculation.
- **Mix LCS.** Given the diversity in planning approaches and individual preferences, we propose a Mixed Longest Common Subsequence (MIX LCS) metric. In our LCS similarity comparison, we consider a generated plan step to be similar if it matches any step in either reference plan.

**Implementation details.** In our Iterative Verification and Refinement process, the maximum iterations is set to 3. In our Retrieval process, we set $k_1$ to 100 and set $k_2$ to 3 or 5 depending on the experimental

---

[5] sentence-transformers/all-MiniLM-L6-v2

Table 3: Comparison of Different Planning method on the Blocks Arrange task.

| Method | Top-K | Average SR |
|---|---|---|
| SayCan [71] | - | 26% |
| GPT-4o w/ COT | - | 33% |
| GPT-4o w/ HP-KG | Top-3 | **45%** |
| | Top-5 | 42% |

Table 4: Comparison of multimodal large models of different scales on Actplan-1K.

| Model | HP-KG | Top-K | LCS ↑ | Mix LCS ↑ | Average ↑ |
|---|---|---|---|---|---|
| GPT-4o-2024-11-20 [54] | - | - | 11.7848 | 12.2700 | 12.0274 |
| | ✓ | Top-3 | 11.8987 | 12.5063 | 12.2025(**+1.45%**) |
| | ✓ | Top-5 | 12.2616 | 12.7552 | 12.5084(**+3.99%**) |
| Gemini-2.0-Flash [55] | - | - | 10.4261 | 11.0717 | 10.7486 |
| | ✓ | Top-3 | 11.0210 | 11.5316 | 11.2763(**+4.90%**) |
| | ✓ | Top-5 | 10.9746 | 11.4767 | 11.2256(**+4.43%**) |
| Qwen2-VL-7B-Instruct [56] | - | - | 7.2909 | 7.8045 | 7.5477 |
| | ✓ | Top-3 | 8.2067 | 9.5527 | 8.8797(**+17.64%**) |
| | ✓ | Top-5 | 8.4092 | 9.0084 | 8.7088(**+15.38%**) |
| Qwen2-VL-72B-Instruct [56] | - | - | 8.5189 | 9.0464 | 8.7826 |
| | ✓ | Top-3 | 9.5527 | 10.0759 | 9.8143(**+11.74%**) |
| | ✓ | Top-5 | 9.2784 | 9.7637 | 9.5210(**+8.41%**) |

Table 5: Comparison of Different Graph Structures on the Actplan-1K. All methods use Top-K=3.

| Graph Structure | LCS | Mix LCS | Average |
|---|---|---|---|
| Qwen2-VL-72B-Instruct [56] | | | |
| - | 8.5189 | 9.0464 | 8.7826 |
| Unstructured Docs | 8.9118 | 9.4345 | 9.1771 |
| Coarse Graph Structure | 8.9113 | 9.2953 | 9.1033 |
| Hierarchical Graph (ours) | 9.5527 | 10.0759 | **9.8143** |

Table 6: Effectiveness analysis of each stage in our automated graph construction framework on Actplan-1K.

| Procedures Generator | Iterative Verification | Summary Agent | Average LCS |
|---|---|---|---|
| × | × | × | 8.7826 |
| ✓ | × | × | 9.4809 |
| ✓ | ✓ | × | 9.5968 |
| ✓ | ✓ | ✓ | **9.8143** |

conditions. For zero-shot manipulation approaches, we utilize GPT-4o as the primary planner unless explicitly stated in experiments. Appendix A provides comprehensive implementation details.

## 4.2 Main Results

**Results on RLBench.** Table 1 presents a comprehensive performance comparison of whether the method is enhanced with our proposed HP-KG. Compared to Voxposer and MA, our HP-KG significantly improves their success rates by up to 10% and 11%, respectively. For all baselines, we achieve improvements on more than half of the tasks, while the remaining tasks are not completed due to the limitations of the baselines' manipulation capabilities. For instance, we find that MA struggles with tasks requiring precise operations due to its lack of precision in identifying target positions, which prevents it from accurately grasping the lid handle during lid-removal tasks. Nevertheless, our HP-KG still shows promising improvements in handling these challenging scenarios. We also compare our HP-KG with COT method, and results are presented in Table 2.

**Results on Blocks Arrange.** Table 3 compares the task success rates of different methods, with GPT-4o serving as the base LLM for all approaches. The results demonstrate that LLMs enhanced with our HP-KG achieve higher success rates compared to both SayCan and COT methods. This performance gain stems from our approach's ability to inject explicit procedural knowledge into LLMs, whereas SayCan and COT methods merely leverage the implicit knowledge embedded in LLMs.

**Results on ActPlan-1K.** Table 4 presents a comprehensive comparison of planning abilities across multi-modal large models with/without HP-KG enhancement. Our HP-KG improves the planning abilities of models across all scales. Specifically, our approach significantly boosts

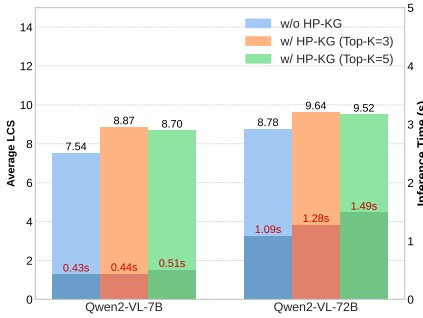

Figure 4: Performance and inference time comparison across different model sizes.

the average LCS of the small-scale model (Qwen2-VL-7B) by 17.64% compared to the 7B baseline, even surpassing the performance of the 72B model baseline. Furthermore, for large-scale models with high capability, such as GPT-4o, Gemini-2.0-Flash, and Qwen2-VL-72B, our approach achieves performance improvements of up to 3.99%, 4.90%, 11.74%, respectively. Comparing results of different retrieval number (top-3 v.s. top-5), we find that the performance of smaller models decreases as the Top-K increases, due to their limited long context comprehension ability. In contrast, for larger models like GPT-4o, the additional procedural knowledge further enhances their performance.

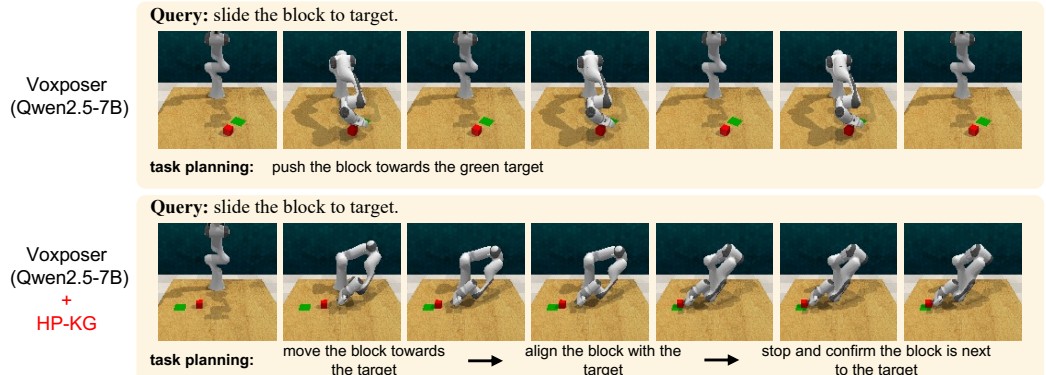

Figure 5: Qualitative Results: Voxposer with small LLM (Qwen2.5-7B) correctly decomposes tasks and executes them successfully when enhanced with our HP-KG framework.

## 4.3 Ablation Studies

**Ablation Study on Hierarchical Structure Design.** To assess the effectiveness of our hierarchical structure, we transform retrieved nodes into various alternative structures and evaluate each variant on the Actplan-1K. Specifically, we introduce two variants: (1) Unstructured Docs, where we directly convert the retrieved nodes into text without sub-graphs; and (2) Coarse Graph Structure, where we extract sub-steps for each retrieved task node and only transform these tasks and their steps into textual descriptions without actions. Table 5 demonstrates that our Hierarchical Graph Structure shows promising improvements over two variants. We find that providing procedural knowledge enhances LLMs' planning capabilities by supplementing limited procedural commonsense. Our hierarchical structure is more effective at organizing procedural knowledge than alternative variants.

**Ablation Study on Graph Construction.** We perform ablation experiments across the three stages in our Automatic Graph Construction Framework. We evaluate Qwen2-VL-72B-Instruct enhanced with HP-KGs from each stage on Actplan-1K, and report Average LCS metric. Table 6 shows that our preliminary procedures graph provides modest improvement over the baseline. Both our Iterative Verification and Refinement process and Summary Agent further enhance the quality of the procedural graph, leading to additional improvements in overall performance.

**Efficiency Analysis.** We also compare the inference time across MLLMs of various scales to evaluate the efficiency of our HP-KG. We deploy Qwen2-VL-7B-Instruct and Qwen2-VL-72B-Instruct on the same hardware platform and evaluate them on the Actplan-1K benchmark, measuring both average inference time and Average LCS metric. Results are presented in Figure 4. As the retrieval number increases (from top-3 to top-5), inference time of LLMs increases slightly due to longer context length. Notably, Qwen2-VL-7B enhanced with our HP-KG achieved performance comparable to the Qwen2-VL-72B baseline while saving 50% of inference time, demonstrating that our method can effectively reduce reliance on large-scale LLMs. We also compare different LLMs within Voxposer for robotics manipulation tasks and evaluate their computational efficiency in Appendix C.

## 4.4 Case Studies

Figure 5 presents a qualitative analysis by comparing scenarios with and without HP-KG enhancement. We utilize Qwen2.5-7B as the planner in Voxposer to decompose tasks into different subtasks and execute them sequentially. The Qwen2.5-7B model alone struggles with proper task decomposition, hindering the robotic arm's ability to complete assigned tasks. In contrast, with HP-KG enhancement, the model generates more logical subtasks, enabling successful execution of the overall task objective. We also conduct experiments in a long-horizon robot task planning benchmark using the Kitchen-World environment [67]. Due to space constraints, detailed results are presented in Appendix D.

## 5 Limitations and Conclusion

In this work, we propose Hierarchical Procedural Knowledge Graphs (HP-KG) to advance long-horizon planning capabilities of LLMs. We introduce a hierarchical graph structure that captures

complex real-world task relationships. We also develop an automatic knowledge graph construction framework powered by LLM-based multi-agents. Furthermore, we propose a retrieval approach to leverage the constructed HP-KG enhancing LLMs for robotic manipulation. Extensive experiments on multiple benchmarks demonstrate our improvements compared to existing approaches. However, our knowledge graph is limited to household activities, constraining its applicability in general scenarios. We plan to develop a general procedural graph and apply it to wider field in the future.

## Acknowledgments and Disclosure of Funding

This work was supported by the National Natural Science Foundation of China (NSFC) under Grant No. 62125201, U24B20174, No. 62306273. The authors declare no competing interests.

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

## A  Overall Experimental Details

**Software and Hardware Configurations.** We conduct our experiments on servers equipped with NVIDIA A6000 GPUs (48GB VRAM), with NVIDIA CUDA Toolkit version 11.8. For inference time comparison, we deploy different-sized models (7B and 72B) using the vLLM inference framework and use the AWQ quantized version for the 72B model to fit within the available GPU memory.

**Experimental Parameter Settings.** In our Iterative Verification and Refinement process, the maximum iterations is set to 3. For all clustering operations, we employ cosine similarity with a threshold of 0.85. In our Procedural Graph Retrieval process, we set $k_1$ to 100 and set $k_2$ to 3 or 5 depending on the experimental conditions. For zero-shot manipulation approaches, we utilize GPT-4o as the primary planner unless explicitly stated in experiments and we set the target retrieval level $L_{\text{target}}$ to the step procedure. For experiments on Actplan-1K, we set the target retrieval level $L_{\text{target}}$ to the Task procedure due to its more complex task objectives.

**Details of the Constructed Graph.** Our HP-KG encompasses a rich variety of actions and objects, enabling comprehensive coverage of diverse tasks in common daily scenarios. Additionally, we present the task distribution of our HP-KG, demonstrating its potential for assisting robots in complex task completion. The specific details are shown below:

Table 7: Statistics of nodes and their corresponding average number of child nodes.

| Node Type | Node Count | Average Sub-Nodes |
|---|---|---|
| Task Nodes | 6380 | 6.58 |
| Step Nodes | 40659 | 4.11 |
| Action Nodes | 136943 | - |

- **Graph Statistics.** Our HP-KG comprises 6K Task Nodes, 40K Step Nodes, and 136K Action Nodes in total, with comprehensive statistics detailed in Table 7. Furthermore, HP-KG incorporates over 1,300 distinct verbs and approximately 26,275 objects, whose distributions are illustrated through word cloud visualizations in Figure 6.
- **Task Length Distribution.** We present the distribution of total task lengths in HP-KG in Figure 7. Our tasks span from simple procedures (approximately 3-7 actions) to complex operations (over 30 actions). This diversity highlights HP-KG's capability to support robotic systems in executing both straightforward and complex task sequences.

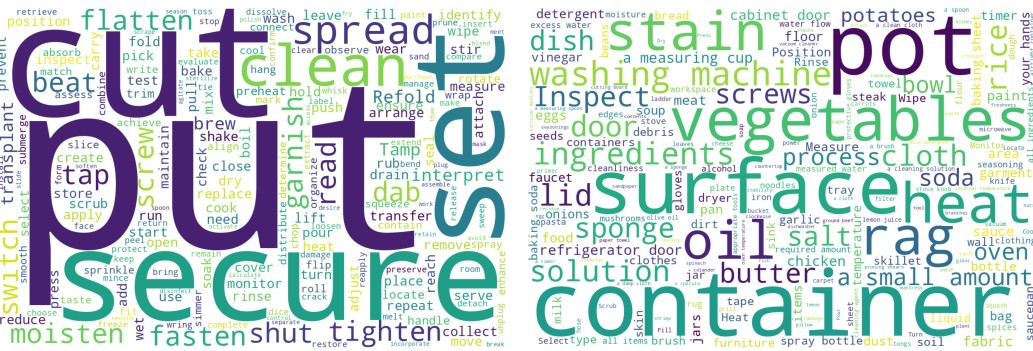

Figure 6: Statistical analysis of verbs and objects in HP-KG actions. **(Left)** Word cloud visualization of verbs. **(Right)** Word cloud visualization of objects.

## B  Procedural Graph Retrieval-Augmented Planning Algorithm

The complete formulation of the retrieval algorithm is detailed in Algorithm 1.

## C  Additional Experiments Results

In this section, we present additional experimental results on both ActPlan-1K and RLBench benchmarks.

**Additional comparison on ActPlan-1K.** Table 8 presents a broader comparison of diverse large vision-language models (e.g., InternVL-2.5-26B [74], Llama-3.2-90B-Vision-Instruct [75]) on the

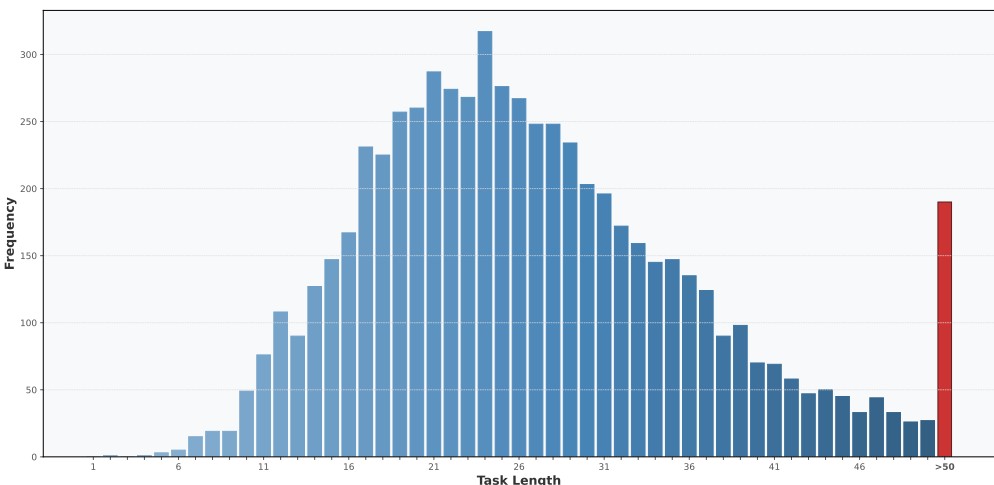

Figure 7: Distribution of total task lengths in HP-KG.

---

**Algorithm 1** Procedural Knowledge Graph Retrieval and Planning.

---

**Input:** General instruction, hierarchical procedural knowledge graph $\mathcal{G} = (\mathcal{V}, \mathcal{E}, X)$, target level $L_{\text{target}}$, parameters $k_1$, $k_2$, hop limit $K$, encoder model
**Output:** Retrieved procedural knowledge in textual format

1: **Query Retrieval:**
2: Extract task objectives as query $x_q$ using LLM
3: Encode query: $z_q = \text{Encoder}(x_q) \in \mathbb{R}^d$
4: Retrieve initial candidates: $V_{k_1} = \text{TopK}_{n \in \mathcal{V}} \text{sim}(z_q, z_n)$ ▷ Top-$k_1$ semantically similar nodes
5: **Entity Expansion and Re-Ranking:**
6: Expand retrieved nodes: $V'_{k_1} = \bigcup_{n \in V_{k_1}} \{n' \mid \text{dist}(n', n) \leq K, n' \in \mathcal{V}\}$ ▷ $K$-hop expansion
7: Filter by target level: $V_{\text{target}} = \{n \in V'_{k_1} \mid \text{Level}(n) = L_{\text{target}}\}$
8: Re-rank candidates: $V_{k_2} = \text{TopK}_{n \in V_{\text{target}}} \text{sim}(z_q, z_n)$ ▷ Final top-$k_2$ nodes
9: **Procedural Text Formatting:**
10: Text $\leftarrow \emptyset$ ▷ Initialize formatted text collection
11: **for** each node $n \in V_{k_2}$ **do**
12:    Retrieve attributes and child nodes of $n$
13:    Format node information into structured text segment $t_n$
14:    Text $\leftarrow$ Text $\cup \{t_n\}$
15: **end for**
16: Concatenate all text segments in Text into final textual context $T$
17: **Return:** $T$ ▷ Formatted procedural text

---

ActPlan-1K [18] benchmark. Since some models cannot fully adhere to instructions and produce complete planning outputs, we record the count of successfully generated planning solutions for each model. Our HP-KG approach demonstrates consistent effectiveness across models of varying scales. We present detailed ablation results of our construction framework in Table 9.

**Additional comparison on RLBench.** In addition to GPT-4o, we integrate several open-source models (Qwen2.5-7B-Instruct, Qwen2.5-72B-Instruct [76]) into Voxposer and conduct efficiency analysis. The success rates across different tasks and inference time comparisons are presented in Table 10 and Figure 8, respectively. Compared to these models without HP-KG, our approach improves their success rates by up to 10.83% and 5.83%, respectively. Furthermore, with HP-KG enhancement, Qwen2.5-7B-Instruct achieves performance comparable to Qwen2.5-72B-Instruct while requiring less than 20% of its inference time.

**Detailed Inference Time Analysis.** We conduct experiments to evaluate the efficiency of our retrieval method. Specifically, we measure the average Instruction Encode Time (Encode Time),Retrieval Time, K-hop Time, Reranking Time, Graph Conversion Time and LLM Inference Time during

Table 8: We evaluated additional large multimodal language models on the ActPlan-1K benchmark. In our analysis, the Success metric indicates the number of instances where models successfully generated planning solutions in the required format.

| Model | HP-KG | Top-K | Success | LCS ↑ | Mix LCS ↑ | Average ↑ |
|---|---|---|---|---|---|---|
| InternVL-2.5-26B [74] | - | - | 237/237 | 8.4430 | 8.6920 | 8.5675 |
| | ✓ | Top-3 | 237/237 | 8.5147 | 8.9915 | 8.7531(**+2.16%**) |
| | ✓ | Top-5 | 237/237 | 8.4599 | 8.9957 | 8.7278(**+1.87%**) |
| Llama-3.2-90B-Vision-Instruct [75] | - | - | 197/237 | 10.3756 | 10.9086 | 10.6421 |
| | ✓ | Top-3 | 204/237 | 11.6715 | 12.4166 | 12.0440(**+13.17%**) |
| | ✓ | Top-5 | 181/237 | 10.8563 | 11.1602 | 11.0082(**+3.44%**) |

Table 9: Detailed results of each stage in our automated graph construction framework on Actplan-1K.

| Procedures Generator | Iterative Verification and Refinement | Summary Agent | LCS | Mix LCS | Average LCS |
|---|---|---|---|---|---|
| × | × | × | 8.5189 | 9.0464 | 8.7826 |
| ✓ | × | × | 9.2320 | 9.7299 | 9.4809 |
| ✓ | ✓ | × | 9.2953 | 9.9578 | 9.6265 |
| ✓ | ✓ | ✓ | 9.5527 | 10.0759 | 9.8143 |

the Top-3 and Top-5 search processes on the Actplan-1K[18]. All settings are consistent with the Experimental Details in Appendix A. Table 11 presents the detailed timing breakdown. Our retrieval pipeline is highly efficient, with the combined overhead (encoding, retrieval, k-hop, reranking, and graph conversion) being significantly smaller than the MLLM inference time.

**Additional experiments on complex manipulation benchmark.** We conduct experiments on a more complex benchmark, VLABench[77], which contains multiple complex tasks involving common sense knowledge, physical rules, and reasoning capabilities. VLABench is designed to evaluate not only VLAs but also MLLMs. In the MLLM evaluation setting, models must output both the skills to be invoked and their corresponding parameters.

We select four task categories: CommonSense, Complex, M&T, and PhysicsLaw, with each category containing 5-8 tasks. The CommonSense, M&T, and PhysicsLaw categories evaluate the model's understanding of world procedural knowledge, while the Complex category requires not only world knowledge but also long-horizon planning capabilities.

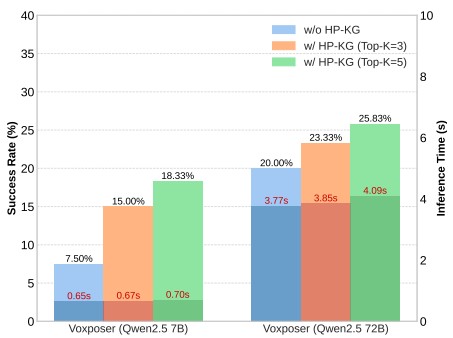

Figure 8: Performance and inference time comparison of different model sizes in the Voxposer framework.

We compare the accuracy of MLLMs enhanced with our HP-KG against those without such enhancement, with each task evaluated over 50 trials. As shown in Table 12, our HP-KG consistently improves planning performance even on complex manipulation benchmark. This indicates that our approach effectively enhances MLLMs' planning capabilities by providing structured prior knowledge that guides decision-making in challenging scenarios.

Table 10: Additional comparison of task success rates (%) for zero-shot robotic manipulation on RLBench using Voxposer [9]. We integrated various open-source large language models into the Voxposer framework.

| Models | HP-KG | Top-K | Open_wine | Take_lid | Take_scale | Take_umbrella | Slide_block | Play_jenga | Average |
|---|---|---|---|---|---|---|---|---|---|
| Voxposer (Qwen2.5-7B-Instruct [76]) | - | - | 15.00% | 0.00% | 5.00% | 10.00% | 15.00% | 0.00% | 7.50% |
| | ✓ | Top-3 | 25.00% | 0.00% | 15.00% | 5.00% | 45.00% | 0.00% | 15.00%(**+7.50%**) |
| | ✓ | Top-5 | 30.00% | 0.00% | 25.00% | 20.00% | 35.00% | 0.00% | 18.33%(**+10.83%**) |
| Voxposer (Qwen2.5-72B-Instruct [76]) | - | - | 5.00% | 0.00% | 25.00% | 55.00% | 35.00% | 0.00% | 20.00% |
| | ✓ | Top-3 | 30.00% | 0.00% | 25.00% | 65.00% | 20.00% | 0.00% | 23.33%(**+3.33%**) |
| | ✓ | Top-5 | 35.00% | 0.00% | 35.00% | 65.00% | 20.00% | 0.00% | 25.83%(**+5.83%**) |

Table 11: Detailed Inference Time Analysis with Qwen2-VL-72B-Instruct on ActPlan.

| Top-k | Encode Time | Retrieval Time | K-hop Time | Reranking Time | Graph Conv. Time | LLM Inference Time |
|-------|-------------|----------------|------------|----------------|------------------|--------------------|
| Top-3 | 0.038s | 0.045s | 0.001s | 0.025s | 0.001s | 1.28s |
| Top-5 | 0.039s | 0.046s | 0.001s | 0.027s | 0.001s | 1.49s |

Table 12: Performance Comparison with Different Top-k Settings on VLABench

| Models | HP-KG | Top-k | CommonSense | M&T | PhysicsLaw | Complex | Average |
|--------|-------|-------|-------------|-----|------------|---------|---------|
| | - | - | 26.15 | 28.19 | 10.21 | 19.01 | 20.89 |
| Intern3-VL-8B [78] | ✓ | 3 | 28.70 | 29.21 | 18.95 | 21.47 | 24.58 |
| | ✓ | 5 | 29.55 | 28.70 | 18.22 | 21.16 | 24.41 |

# D   Long-Horizon Planning Experiments

We further investigate the capability of our HP-KG to support embodied robots in long-horizon planning tasks by conducting evaluations in the more sophisticated simulated environment Kitchen World [67].

**Kitchen World Settings.** The environment comprises either a single-arm or dual-arm robot and generated kitchen scenarios containing multiple movable objects and articulated objects or surfaces. The robot is required to complete tasks based on given natural language instructions. Following Yang et al. [67], we consider a task where a single-arm robot is required to make chicken soup in this kitchen environment. It is worth noting that this represents a highly complex long-horizon task, requiring approximately a dozen sequential actions for successful completion. Specifically, we consider a complex scenario where all cabinet doors are initially closed and the pot is covered with its lid.

We adopt GPT-4o [54] as our base model for generating sub-goals and apply the default TAMP approach from Yang et al. [67] to solve the generated sub-goals. Similar to Yang et al. [67], we use Task Completion Percentage as our metric, defined as the proportion of solved subproblems among all subgoals.

**Results.** Table 13 presents a performance comparison between models enhanced with our proposed HP-KG and those without such enhancement. The results demonstrate that base models enhanced with HP-KG achieve higher Task Completion Percentage, indicating that our HP-KG can effectively facilitate robot execution of complex long-horizon tasks. Additionally, we present a qualitative comparison in Figure 9, which illustrates the subgoals generated by different methods during the experiments. GPT-4o enhanced with our HP-KG can generate more reasonable subgoals, while the baseline model without HP-KG may omit certain essential operations (e.g., opening cabinets and picking up salt and pepper).

Table 13: Comparison of Task Completion Percentage for Kitchen Task Execution.

| Method | Top-K | Task Completion Percentage |
|--------|-------|----------------------------|
| GPT-4o [54] | - | 77.4% |
| GPT-4o w/HP-KG | Top-3 | 86.9%(**+9.5%**) |
| GPT-4o w/HP-KG | Top-5 | 83.5%(**+6.1%**) |

# E   Implementation Details of the Automated Graph Construction Framework

**Details of Verification Rules** In our Iterative Verification and Refinement process, we construct a comprehensive set of validation rules to evaluate whether the generated procedures adhere to procedural principles. We categorize these rules based on their application types (e.g., tasks, steps, actions) and present several examples in Table 14.

**Instruction Template in Construction Framework.** We provide the prompt templates used at each stage of our Framework. Specifically, Table 15 and Table 16 present the prompts for procedure generation and completion. Table 17 shows the prompt that guides the model to aggregate rule verification results and generate corresponding feedback and scores, while the prompt in Table 18 instructs the Refiner to modify procedures based on this feedback. Finally, the Summary Agent employs the prompt in Table 19 to merge redundant procedures. In each stage of our framework, we

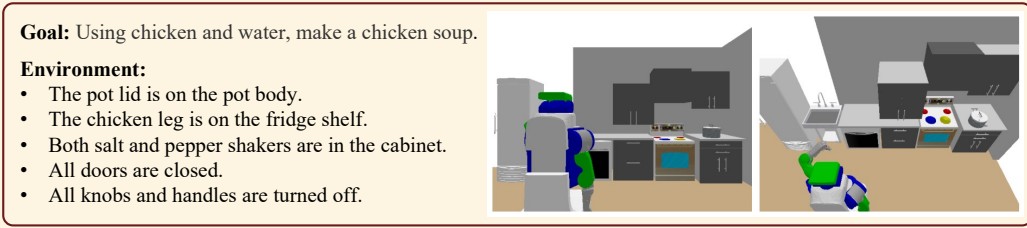

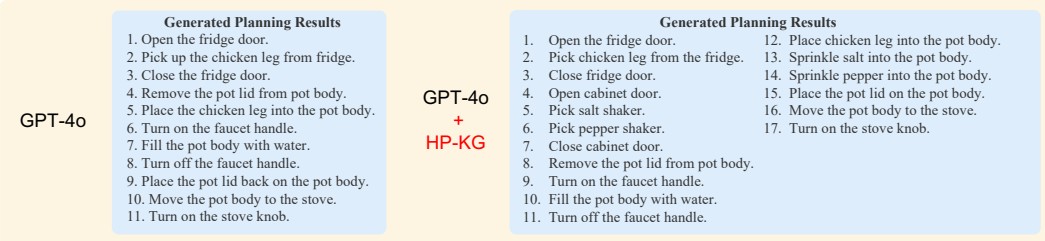

Figure 9: Qualitative Results in the Kitchen World Environment. GPT-4o enhanced with our HP-KG generates more reasonable subgoals, while the baseline model without HP-KG may omit certain essential operations.

Table 14: Validation rules based on established procedural principles

| Procedure Type | Rules |
| --- | --- |
| Task Procedure | Check if task name contains specific and unambiguous descriptors. |
| | Check if task name avoids vague terms. |
| | Check if task name clearly indicates the purpose. |
| Step Procedure | Check if step name avoids ambiguous terms descriptors. |
| | Check if description explains purpose and necessity. |
| | Check if description avoids redundant information. |
| | Check if steps follows logical sequence. |
| Action Procedure | Check if action follows basic structure. |
| | Check if action is atomic and independently executable. |
| | Check if description explains specific goal. |
| | Check if actions follows logical sequence. |
| | Check if actions follows logical sequence. |
| | Check if actions avoids redundancy with other actions. |

employ GPT-4o as our primary LLM to generate outputs based on the prompts. Notably, for verifying procedures' compliance with rules, we utilize Deepseek-V3, which offers comparable performance to GPT-4o but at a lower operational cost.

# F Example of Procedural Graph

We present an example of our HP-KG, as illustrated in Figure 10. This case details the procedural steps and associated actions for completing the task 'clean broom and remove stains'. Each procedure includes three components: a name, detailed description, and practical tips.

Table 15: This designed prompt is used to extract tasks and their corresponding step-by-step instructions from WikiHow documents.

Analyze multiple similar Wikihow examples to extract canonical procedural knowledge.

Input:
{examples}

Thinking Process:
1. Compare all examples to identify:
- Common goal across all tasks
- Essential steps present in examples
- Conflicting instructions (resolve via majority vote)

2. For each step:
- Find the most precise action verb used
- Note critical tips from all variants

Output in the following JSON format:
{
    "task": "[specific purpose/goal] [main how-to]",
    "description": "Contextual summary containing: [Operational significance and scope boundaries, Key differentiators from alternative approaches, Optimal application scenarios]",
    "tips": "Comprehensive list of potential issues, important tips, and warnings",
    "steps": [{
        "step": "[PrecisionVerb] [TargetObject] [specific purpose]",
        "description": "Detailed step description including: [ What actions to take, Why these actions are necessary, Expected outcomes, Important considerations or precautions"],
        "tips": "Step-specific considerations, timing requirements, measurements, or safety precautions"
}]
}
Please ensure your analysis:
- Includes safety considerations and edge cases
- Provides clear, actionable steps
- Lists relevant tips and warnings for each step
- Provides generalized solutions that can be adapted to various scenarios
- Considers different environmental conditions and constraints that might affect the task execution

Naming and Description Requirements:

1. Task Names:
- Follow the structure: "[specific purpose/goal] [main purpose]"
- Use clear, specific, and unambiguous descriptions
- Clearly indicate the purpose

2. Task Descriptions:
- Explain the purpose and importance
- Include task-specific tips
- Highlight key warnings or special requirements

3. Step Names:
- Follow a logical sequence
- Avoid ambiguous terms
- Focus on what to do, not why or how
- Ensure logical flow between steps

4. Step Descriptions:
- Detail specific actions and their purposes
- Explain why each action is necessary
- Describe expected outcomes
- Provide step-specific tips and precautions

Table 16: This action completion prompt facilitates the generation of specific actions.

I will provide a complete task breakdown with the following components:
Task Objective: {task}
Task Description: {description}
Important Tips and Considerations: {tips}
Detailed Steps: {steps}
Please Generate Action For All Steps.

Requirements for Action Generation:
1. Action Requirements
   Action sentences structure: [Action] [Object] [Spatial: Object] [Orientation] [Direction][State]

2. Required Information for Each Action:
   - Description: Detailed explanation including specific goal of this atomic action, Why it's necessary at this step, Required preconditions, Expected outcome/success criteria.
   - Tips: List of specific considerations: Safety precautions, Quality control points, Common mistakes to avoid, Environmental considerations.

3. Anti-Redundancy Guidelines:
   A. Consolidate Actions When: Identical action verbs on similar object types, Same execution method/tool/pattern, Order is flexible/interchangeable, Common success metrics apply, Single verification works for all items.
   B. Keep Separate When: 1. Different tools/methods per item, 2. Strict sequence dependencies exist, 3. Unique safety protocols per item, 4. Individual verification required, 5. Different quality standards apply.

Please, think through this step:
1. What is the goal of this step?
2. What are the key constraints and requirements?
3. What safety factors need consideration?
4. What tools or resources are needed?

Then, break down the step into atomic actions, finally provide your response in this JSON format:
{
     "action": "[Action:verbnet verb] [Object:wordnet noun] ...",
     "tips": "Key safety and execution tips",
     "description": "Why this action and what to expect"
}

Table 17: This prompt is used for rule-based validation, compiling different rules results and generated procedures, then prompting the language model to produce comprehensive feedback and score.

Given validation results from rule checks, please analyze the findings and generate improvement recommendations.

Input:
     Original Prompt: {prompt}
     Generated Response: {response}
     Rule Check Results: {rules checking results}

Analysis Tasks:
     1. Review each rule validation result
     2. Identify patterns in rule violations
     3. Determine root causes
     4. Provide corrections feedback

Please provide your evaluation in the following JSON format:
{
     "reason": "Detailed analysis explaining the scoring rationale and key observations",
     "score": number
     "feedback": [
          "Clear and actionable improvement suggestions",
          "Each point focusing on a specific aspect to enhance",
          ...
     ]
}

Table 18: This prompt is utilized to optimize generated procedures based on verification results.

Given an original response and multiple feedback points, along with their corresponding refined versions, please synthesize a final refined response.

Original Response:{response}

Feedback Points:{feedbacks}

Rule Cheks: {rules checking results}

Please analyze all refinements and combine their improvements to generate an optimized final response that addresses all feedback points cohesively.

Please return only the final refined response.

Please provide your response refinements in the following JSON format:

```
{
     "reason": "Detailed analysis of why these refinements are suggested"
     "refined": {refine result}
}
```

Table 19: We employ this prompt to synthesize multiple candidate procedures into a consolidated summary.

Given multiple semantically similar items, analyze and synthesize them into a single, comprehensive and actionable summary.

Input:
{candidate list}

Follow these steps to analyze and summarize the items:

1. Semantic Analysis:
    - Identify core actions and objectives in each item
    - Find common elements and key differences
    - Extract the essential meaning and purpose

2. Content Synthesis:
    - Use specific verbs and nouns, avoid abstract terms.
    - Preserve all critical operational steps and important details.
    - Distill redundant information while keeping context-specific requirements.
    - Synthesize similar concepts into unified expressions.

3. Structured Output Requirements:
    - Title: Use concrete verbs that reflect the main task, include both the action and the target object, keep it brief but specific, make it immediately understandable.

    -Description: Start with the concrete purpose and scope, detail specific steps or methods, include key conditions or requirements, explain the expected outcome, use precise, actionable language.

    -Tips: List specific, executable actions, include crucial warnings or requirements, focus on practical guidance, avoid vague suggestions, synthesize similar tips into stronger, unified points.
4. Quality Verification:
    - Ensure it capture all essential information from original items.
    - Ensure all instructions specific and actionable.
    - Ensure all content logically organized.
    - Ensure the language clear and concrete.

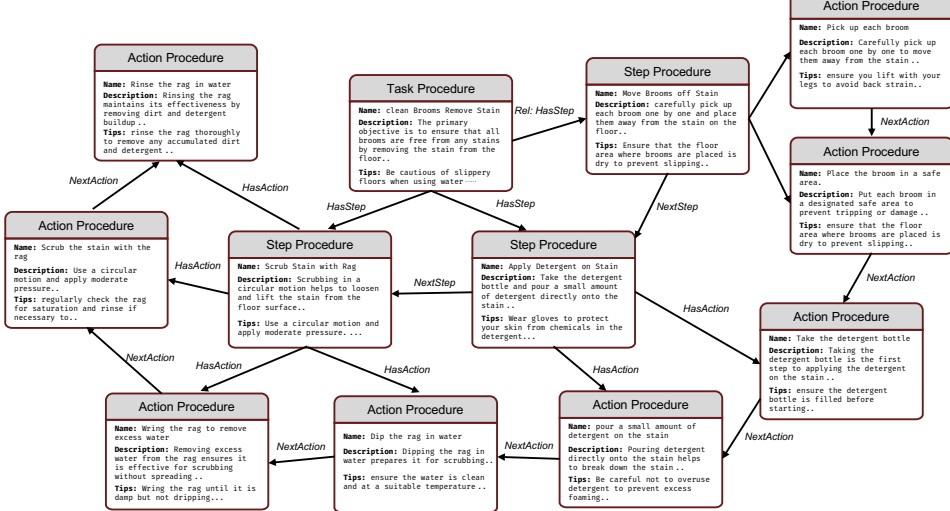

Figure 10: An example of cleaning and removing stains from brooms in our procedural knowledge graph.

