# OpenReview forum: "Enhancing LLM Planning for Robotics Manipulation through Hierarchical Procedural Knowledge Graphs"
_NeurIPS.cc/2025/Conference — NeurIPS 2025 poster_

### Official Review · Reviewer_VoF5 · 2025-06-27

**Clarity:** 2
**Significance:** 2
**Originality:** 3
**Rating:** 3
**Confidence:** 2

**Summary:**

The paper proposes HP-KG, a large-scale hierarchical procedural knowledge graph designed to enhance LLM-based planning for complex manipulation tasks while allowing the use of smaller LLMs.

**Questions:**

- Could you provide automatic or human-annotated precision/coverage statistics (e.g., % of actions truly atomic, % of step ordering correct) for HP-KG?

- Do you foresee the same structure and pipeline scaling to industrial assembly or navigation tasks? Any preliminary results or challenges?

- Also, I would encourage the trials on more challenging manipulation benchmarks like RoboCerebra.

- Please report average retrieval + sub-graph conversion time and memory footprint for top-3/top-5 on ActPlan-1K, and discuss impact on total inference.

- To mitigate potential bias from GPT-4o judging, could you include a small human study validating that HP-KG plans are indeed more feasible/executable?

- The KG Retrieval reminds me the the in-context prompts used in CaP or Instruct2Act, can the authors provide more discussions.

- Overall, due to the limited task complexity in the chosen RLBench, it did not fully demonstrate the effectiveness of the promised KG.

**Ethical Concerns:**

["NO or VERY MINOR ethics concerns only"]

**Final Justification:**

Overall, I think this is a good paper in the view of agent-style VLA to enhance its reasoning or planning ability. My main concern still holds, I thought it would greatly enhance the paper's strength when the authors demonstrated the effectiveness where the proposed KG would further boost the performance of the current SOTA VLA like Pi-0 on the large-scale and more challenging benchmarks, like LIBERO-10 and others.

In view of this, I am currently holding my current score, but would like to lower my confidence.

**Limitations:**

yes

**Quality:**

3

**Strengths And Weaknesses:**

## Strengths

- The overall method is well-motivated where hierarchical KG that directly aligns language instructions with executable robot actions, filling the abstraction gap.
- Fully automated construction via LLM agents with rule feedback reduces manual effort and produces a sizable, publicly promised dataset.
- Model-agnostic enhancement – works with small (7 B) and large (GPT-4o) planners, yielding both quality gains and inference-time savings.

## Weakness

 -  focuses on table-top tasks; transfer to industrial or other domains untested.

-  although clustering/verification exists, no quantitative quality metrics (precision/consistency) for the final KG are reported.

-  author mentioned "safety-critica"  in Line 125, but no further information provided.

- embedding + K-hop search latency on large graphs is not profiled

---

> ### Author Rebuttal · Authors · 2025-07-31
>
> - **W1: Focuses on table-top tasks; transfer to industrial or other domains untested.**
>
> It is possible that our pipeline could be adapted to other domains such as industrial assembly. Our core methodology may be applicable provided that suitable large-scale datasets are available for these domains. However, due to time constraints during the rebuttal period, we do not have sufficient time to construct new large-scale knowledge graphs, as this process requires substantial data collection and curation efforts. As mentioned in our limitations, the study is limited to household activities. Different domains would likely require domain-specific adaptations in entity types, relationship categories, etc.
>
>
> - **W2: No quantitative quality metrics (precision/consistency) for the final KG are reported.**
>
> We conduct experiments to verify the quality of our generated data. Specifically, we sample a subset(100task and 864 steps) of our HP-KG. We use three data quality metrics, including truly atomic actions precision (AAP), step ordering precision (SOP), and actions ordering precision (AOP).
>
> To measure the atomic actions precision, we extracted all verbs from the HP-KG and manually created an atomic verb dictionary containing only atomic verbs (e.g., pick, turn on) while excluding non-atomic ones (e.g., arrange). We then calculated the percentage of atomic verbs in the generated actions based on this dictionary. For Steps Precision and Action Precision, we annotated the correct steps/actions ording and calculate the average percentage of correctly ordered steps and actions, respectively. Detailed results are presented in following Table. Experiments show that our LLM Agents can generate reasonable atomic actions and acceptable steps, helping to bridge high-level human instructions with atomic actions that robots can understand.
>
> ||AAP |SOP|AOP|
> |-|-|-|-|
> |HP-KG|82.39%|86.92%|85.78%|
>
> To better illustrate our HP-KG approach, we include a graph example in Figure 5 and detailed graph statistics in the Appendix of the Supplementary Material. We hope these supplementary materials may offer helpful context for understanding our HP-KG.
>
>
> - **W3: author mentioned "safety-critica" in Line 125, but no further information provided.**
>
> The term 'safety-critical' refers to the 'safety-critical guidelines' mentioned in Line 125. These are essential safety warnings and precautionary instructions that our procedure should incorporate to alert robot to potential accidents during procedure execution. Following this principle, we incorporated a Tips attribute when defining the properties of the Procedure. An example of a Tip is: "Handle the notebook carefully to avoid bending or tearing pages.
>
> - **W4: embedding + K-hop search latency on large graphs is not profiled.**
>
> We conduct experiments to evaluate the efficiency of our retrieval method. Specifically, we measure the average Instruction Encode Time (Encode Time),Retrieval Time, K-hop Time, Reranking Time, Graph Conversion Time and LLM  Inference Time during the Top-3 and Top-5 search processes on the Actplan-1K[1].
>
> The results are presented in the following table.
>
> |Top-k|Encode Time |Retrieval Time|K-hop time|Reranking Time|Graph Conversion Time|LLM Inference Time |
> |-|-|-|-|-|-|-|
> |Top-3|0.038s|0.045s|0.001s|0.025s|0.001s|1.28s|
> |Top-5|0.039s|0.046s|0.001s|0.027s|0.001s|1.49s|
>
> Results show that our retrieval process is highly efficient, with only minimal additional time overhead that is significantly less than MLLM inference time.
>
>
> - **Q1: Could you provide automatic or human-annotated precision/coverage statistics (e.g., % of actions truly atomic, % of step ordering correct) for HP-KG?.**
>
> Thank you for your valuable comment. We have conducted experiments to evaluate the precision and coverage statistics of HP-KG. Detailed results can be found in **W2**
>
>
> - **Q2: Do you foresee the same structure and pipeline scaling to industrial assembly or navigation tasks? Any preliminary results or challenges?**
>
> It is possible that our pipeline could be adapted to other domains such as industrial assembly or navigation tasks. The core methodology may be applicable provided that suitable large-scale datasets are available for these domains. However, due to time constraints during the rebuttal period, we do not have sufficient time to construct new large-scale knowledge graphs, as this process requires substantial data collection and curation efforts.
>
> As mentioned in our limitations, the study is limited to household activities. Different domains would likely require domain-specific adaptations in entity types, relationship categories, etc.
>
>
> - **Q3: Also, I would encourage the trials on more challenging manipulation benchmarks like RoboCerebra.**
>
> Thank you for your insight. We further conducted experiments on a more complex benchmark. Since the RoboCerebra benchmark is not open source, we conducted our experiments on VLABench[2].
>
> VLABench, which contains multiple complex tasks that involve common sense knowledge, physical rules, and reasoning capabilities. VLABench can evaluate not only VLAs but also MLLMs. In the MLLM evaluation setting, MLLMs are required to output the skills to be invoked and the corresponding parameters for each skill.
>
> We select four task categories (CommonSense, Complex, M&T, PhysicsLaw), with each category containing 5-8 tasks. CommonSense, M&T, and PhysicsLaw categories evaluate the model's understanding of world procedural knowledge, while Complex categoriy, beyond world knowledge, demand long-horizon planning. We compare the accuracy of MLLMs enhanced with our HP-KG against those without such enhancement, with each task run for 50 trials. Results are presentsed on following Table.
>
> |Models|Top-k|CommonSense|M&T|PhysicsLaw|Complex|
> |-|-|-|-|-|-|
> |Intern3-VL-8B|-|26.15 |28.19 |10.21||
> |Intern3-VL-8B + HP-KG |3|28.70|29.21|18.95|
> |Intern3-VL-8B + HP-KG |5|29.55|28.7|18.22|
>
> Results demonstrate that our HP-KG can enhance MLLMs' planning capabilities by injecting procedural commonsense into the LLM context, thereby providing structured prior knowledge that facilitates robust planning in complex tasks.
>
>
> - **Q4: Please report average retrieval + sub-graph conversion time and memory footprint for top-3/top-5 on ActPlan-1K, and discuss impact on total inference.**
>
> Thank you for your valuable comment. We have conducted experiments to evaluate the time performance of the Top-k retrieval process. Detailed results can be found in **W4**. The results demonstrate that our retrieval process is highly efficient, incurring only minimal additional time overhead that is significantly less than MLLM inference time
>
>
> - **Q5: To mitigate potential bias from GPT-4o judging, could you include a small human study validating that HP-KG plans are indeed more feasible/executable?.**
>
> We conduct experiments to verify the feasibility of our plans of HP-KG. Specifically, we sample a subset (100 tasks) from our HP-KG and manually annotate whether each action is feasible or executable for the robot. We then calculate the average percentage of feasible actions, denoted as the feasible action rate (FAR).
>
> Detailed results are presented in following Table. Experiments show that our LLM Agents can generate feasible atomic actions, helping to bridge high-level human instructions with atomic actions that robots can understand.
>
> || FAR|
> |-|-|
> |HP-KG| 71.54%|
>
>
>
> - **Q6: The KG Retrieval reminds me the the in-context prompts used in CaP or Instruct2Act, can the authors provide more discussions**
>
> Instruct2Act presents an agent-based robotic manipulation framework that generates Python code based on task instructions, leverages this code to invoke vision tools (such as SAM) for environmental perception, and subsequently generates plausible robotic actions for low-level controllers to execute.
>
> Our approach is orthogonal to these agent-based robotic manipulation frameworks and provides complementary capabilities. The constructed HP-KG can be seamlessly integrated into various planning generation systems, offering procedural commonsense knowledge to facilitate more rational action planning.
>
>
> - **Q7: Overall, due to the limited task complexity in the chosen RLBench, it did not fully demonstrate the effectiveness of the promised KG.**
>
> We conduct experiments on more complex tasks using VLABench. Detailed experimental results can be found in **Q3**.
>
>
> [1] ActPlan-1K: Benchmarking the Procedural Planning Ability of Visual Language Models in Household Activities. EMNLP 2024.
>
> [2] VLABench: A Large-Scale Benchmark for Language-Conditioned Robotics Manipulation with Long-Horizon Reasoning Tasks. ICCV 2025.

---

> > ### Comment · Reviewer_VoF5 · 2025-08-05
> >
> > Thanks for the author's rebuttal.
> >
> > My concerns on W2, W4, Q3, Q4, Q5, Q6 are almost clear.
> >
> > However, from the original response, I have raised some questions:
> >
> > 1. W1: you mentioned that due to the time limit, you cannot "construct new large-scale knowledge graphs", so my question would be how large the dataset need to be to satisfy the needs for the new domain? Would there be great domain gap? Could there be any possibility to reuse the current one and build a new minor subset?
> >
> > 2. W3: I still did not find any definition on the so-called safety in your main paper.

---

> > > ### Author Response · Authors · 2025-08-08
> > >
> > > Dear Reviewer, we sincerely appreciate your time and effort in evaluating our work. we hope the following responses can address your concerns
> > >
> > > **W1: How large the dataset need to be to satisfy the needs for the new domain? Would there be great domain gap? Could there be any possibility to reuse the current one and build a new minor subset?**
> > >
> > > 1. The size depends on the specific task and the amount of available data. For household tasks, we selected content from how-to databases (one of the largest online databases containing nearly 10k how-to articles across various everyday life scenarios) and filtered out low-quality and duplicate content. The exact dataset size required for the new domain is difficult to quantify, as it depends on both the scope of the domain and the volume of accessible data. We believe the dataset should encompass as much domain knowledge as possible to ensure better generalizability.
> > >
> > > 2. The domain gap depends on how different the tasks are across domains. In our case, since our knowledge graph is developed for household tasks, it may not be applicable to navigation tasks due to fundamental differences in task requirements. For example, household tasks encompass manipulation-based actions (e.g., grasping objects, pressing buttons), while navigation tasks center on movement-related actions such as object tracking and trajectory following.
> > >     We also conduct experiments on navigation tasks. We select Navid [1] as our baseline and evaluate performance on the R2R [2] navigation benchmark, which is one of the most widely recognized benchmarks in Vision-and-Language Navigation (VLN).
> > >
> > >     We follow the standard VLN evaluation metrics to assess navigation performance, including success rate (SR) and navigation error from goal (NE). We compare the performance of Navid with and without our HP-KG enhancement across these metrics. Results are presented in following Table. Experiments demonstrate that directly applying our HP-KG to the navigation domain leads to performance degradation due to task differences between navigation and household tasks. This indicates that particular domains still require tailored knowledge graphs.
> > >     |Method|NE↓|SR↑|
> > >     |-|-|-|
> > >     |Navid|5.6|42.0%|
> > >     |Navid + HP-KG|7.03|37.6%|
> > >
> > > 3. Yes, there's potential for reuse in our current HP-KG. In our hierarchical design, the top-level tasks are domain-specific, while actions and steps at lower levels tend to be more generalizable. For overlapping domains, these lower-level components could be reusable and would only require building a new minor subset. However, for completely unrelated domains, it would be difficult to reuse the existing knowledge graph.
> > >
> > > In summary, Knowledge graphs are generally domain-specific (for instance, a Financial Knowledge Graph[3] performs well within the financial field but is difficult to transfer to unrelated domains). This represents a fundamental limitation of knowledge graphs themselves. Within our household domain, our HP-KG has demonstrated considerable effectiveness, proving the value of our approach.
> > >
> > > **W2: I still did not find any definition on the so-called safety in your main paper.**
> > >
> > > We apologize for any confusion caused by the unclear explanation.
> > > We use the term 'safety-critical guidelines' in Line 125, which is inspired by [4] and properly cited in our manuscript(Line 126). 'Safety-critical guidelines' refer to instructions that help avoid potential mistakes during procedure execution. We implement this by incorporating a Tips attribute into Procedure Nodes. However, we acknowledge that we did not provide a sufficiently clear explanation of this concept in the original manuscript. We will include a detailed definition in the revised version.
> > >
> > >
> > >
> > >
> > >
> > >
> > >
> > > [1] NaVid: Video-based VLM Plans the Next Step for Vision-and-Language Navigation. RSS 2024.
> > >
> > > [2] Beyond the Nav-Graph: Vision-and-Language Navigation in Continuous Environments. ECCV 2020.
> > >
> > > [3] Exploring Large-Scale Financial Knowledge Graph for SMEs Supply Chain Mining. IEEE Trans. Knowl. Data Eng. 36(6)
> > >
> > > [4] PARADISE: Evaluating Implicit Planning Skills of Language Models with Procedural Warnings and Tips Dataset. ACL 2024.

---

> > > > ### Comment · Reviewer_VoF5 · 2025-08-08
> > > >
> > > > Thank the authors for your time and effort.
> > > >
> > > > W1: Current theoretical analysis is good but I would encourage more experimental results in your final / revised version to support your analysis.
> > > >
> > > > W2: Thanks for this information.
> > > >
> > > > Overall, I thought this paper is a good paper from the view of KG and agent-style VLA. My main concerns is still holding as I stated in the first round: How this method contributes to an already good enough VLA models (like Pi0, UniVLA, DexVLA, etc), and how the final performance in the commonly used (for VLA research), like LIBERO and CALVIN.
> > > >
> > > > I would encourage the authors give some initial discussions on this direction, and due to the time limitations, no further experiments are required.
> > > >
> > > > Currently, I would keep my score but lower my confidence. But if the authors could convince me, I would consider raising my score.

---

### Official Review · Reviewer_LNsU · 2025-07-02

**Clarity:** 3
**Significance:** 3
**Originality:** 3
**Rating:** 5
**Confidence:** 3

**Summary:**

This paper introduces Hierarchical Procedural Knowledge Graphs (HP-KG) as a means to enhance the planning capabilities of Large Language Models (LLMs) for complex robotic manipulation tasks, while also aiming to reduce reliance on large-scale models. The authors identify two key limitations in current LLM-driven manipulation systems: (1) a lack of accurate procedural knowledge for complex tasks and (2) the inefficiency and high energy cost of deploying large LLMs in real-world robotic systems.

To address these issues, the proposed HP-KG models the hierarchical structure of tasks — breaking them down into steps and atomic robotic actions — thereby aligning human instructions with machine-executable procedures. The knowledge graphs are automatically constructed using an LLM-based multi-agent system, reducing the need for manual annotation while maintaining quality. The resulting dataset includes over 40,000 activity steps covering more than 6,000 household tasks.

Empirical results suggest that small-scale LLMs (e.g., 7B parameters) augmented with HP-KG outperform significantly larger models (e.g., 72B) in planning tasks, and the benefits also transfer to state-of-the-art models like GPT-4o.

**Questions:**

N/A

**Ethical Concerns:**

["NO or VERY MINOR ethics concerns only"]

**Final Justification:**

Overall, I think it's a good paper. My concerns about the anonymous link and the real robot deployment are not resolved, but I think it's acceptable.

**Limitations:**

Yes

**Quality:**

3

**Strengths And Weaknesses:**

## Summary of Strengths
1.	This paper is well written and clearly organized, making it easy to follow.

2.	The idea of using procedural knowledge graphs to enhance the planning of LLMs is straightforward and experimental results well supported the results.

3.	I like the idea of constructing the knowledge graph from the wikihow dataset, the authors promised to release the knowledge graph, so it would be helpful for the community to build more powerful LLM planning modules for embodied AI.

## Summary of Weaknesses
1.	The anonymous link didn’t provide any code or data for us to review.

2.	The lack of experiments or demos in real-world robots, which could decrease the contribution of the world. Real-world scenarios are more complicate and it would be nice to test the robustness of HP-KG there.

---

> ### Author Rebuttal · Authors · 2025-07-31
>
> - **W1: The anonymous link didn’t provide any code or data for us to review.**
>
> Thank you for your valuable feedback. Due to NeurIPS's policy this year, we cannot update our repository or submit any links during the rebuttal period. We commit to updating our repository and releasing the data immediately after the rebuttal period ends to further facilitate Procedural Knowledge development and research.
>
> To better illustrate our HP-KG approach, we include a graph example in Figure 5 and detailed graph statistics in the Appendix of the Supplementary Material. We hope these supplementary materials may offer helpful context for understanding our HP-KG.
>
> - **W2: The lack of experiments or demos in real-world robots, which could decrease the contribution of the world.**
>
> Thank you for your sincere comment. Due to time and cost constraints, we have not yet deployed our approach to real-world robots. We will further extend our HP-KG to real-world applications in future work.

---

### Official Review · Reviewer_uR6N · 2025-07-02

**Clarity:** 4
**Significance:** 4
**Originality:** 3
**Rating:** 5
**Confidence:** 4

**Summary:**

The paper tackles an important task of making the robot manipulation both flexible thanks to LLM model and following the given procedure by utilizing knowledge graphs. The introduced hierarchical knowledge graphs when integrated with LLM outperform compared former models as well as CoT on most of the selected household tasks in RLbench and Actplan. It significantly advances performance especially of the smaller LLM models.

**Questions:**

1) p.5 l193 feedback, textual results descriptions - what these mean, where they are collected and how are these utilized?
2) how do you make sure not to loose some important rule or information by summary agent? two similar procedures might differ only in one small, but important parameter

3) why in Table 1/2 top 5 sometimes performs worse and sometimes better than top 3? why CoT drops performance in Table 2 for slide block compared to RVT? why top 3/5 sometimes performs even worse than RVT? are there some typical problems?

4) table 1/2 shows mainly pick and place tasks, what about more complex tasks like preparing table or tasks like screwing, wiping, and longer tasks like tidying up a table? could you show results on these or discuss limitations of your approach?

**Ethical Concerns:**

["NO or VERY MINOR ethics concerns only"]

**Limitations:**

Limitations are very sparse, mentioning only extension outside household tasks.

**Paper Formatting Concerns:**

in general very well written
- Fig.4 caption state it is on ActPlan
- p.5 l 198 - what are levels?
- p.5 l 200 - e.g.. one extra dot

**Quality:**

4

**Strengths And Weaknesses:**

Strengths:
The paper is very nicely written, clearly structured and provides several meaningful evaluations including inference time comparison. It also tackles an important issue how to make open vocabulary models more efficient by utilizing contextual knowledge. The proposed hierarchical knowledge graphs are clearly described and meaningfully designed.

Weaknesses:
the paper seem to focus evaluation on pick and place tasks. it is not clear how more complex tasks with different parameters (trajectory to follow, pressure, etc) and dexterous manipulations could be represented and how well it performs.

It should be clearer described in which cases it decreases performance compared to model without it or compared to CoT and why.

the evaluation is comparing different methods in different cases, making it difficult to compare.

---

> ### Author Rebuttal · Authors · 2025-07-31
>
> We appreciate your valuable insights and comprehensive review. Your thoughtful comments will greatly enhance our manuscript.
>
> - **W1: Validation on complex tasks and performance remain unclear.**
>
> we conduct comprehensive experiments on VLABench[1], a more complex benchmark comprising eight distinct manipulation skills (e.g., Insert, Hang, Twist) that extend beyond basic pick-and-place operations. These tasks require comprehensive commonsense knowledge, understanding of physical principles, and sophisticated reasoning capabilities to solve effectively. VLABench can evaluate not only VLAs but also MLLMs. In the MLLM evaluation setting, MLLMs are required to output the skills to be invoked and the corresponding parameters for each skill.
>
> We select four task categories (CommonSense, Complex, M&T, PhysicsLaw), with each category containing 4-8 tasks. CommonSense, M&T, and PhysicsLaw categories evaluate the model's understanding of world procedural knowledge, while Complex categoriy, beyond world knowledge, demand long-horizon planning. We compare the accuracy of MLLMs with and without our HP-KG enhancement, and we run each task for 50 trials. Results are presented in the following table.
>
> |Models|Top-k|CommonSense|M&T|PhysicsLaw|Complex|
> |-|-|-|-|-|-|
> |Intern3-VL-8B|-|26.15 |28.19 |10.21|19.01|
> |Intern3-VL-8B + HP-KG |3|28.70|29.21|18.95|21.47|
> |Intern3-VL-8B + HP-KG |5|29.55|28.7|18.22|21.16|
>
> Results demonstrate that our HP-KG can enhance MLLMs' planning capabilities by injecting procedural commonsense into the LLM context, thereby providing structured prior knowledge that facilitates robust planning in complex tasks.
>
> Additionally, our HP-KG framework is primarily designed for providing procedural commonsense to LLM context and facilitating more reasonable planning, rather than addressing dexterous manipulation benchmarks such as [3,4]. These benchmarks primarily evaluate models' ability to perform precise finger movements, coordinated multi-finger grasping, and object manipulation. Therefore, our method is not suitable for dexterous manipulations task.
>
> - **W2: It should be clearer described in which cases it decreases performance compared to model without it or compared to CoT and why.**
>
> 1. On the Slide Blocks task, our method with top-5 retrieval yields lower performance compared to the approach without HP-KG. The task's ambiguous instructions result in more irrelevant information being retrieved, thereby degrading planning quality.
>
> 2. Our method shows comparable performance to COT on Take_lid and Take_scale tasks. This may be due to the inherent limitations of the RVT model on these two tasks, where neither our method nor COT achieves satisfactory performance.
>
>
> - **W3: the evaluation is comparing different methods in different cases, making it difficult to compare.**
>
> Table 1 and Table 2 compare different types of methods. In Table 1, we present a comparison of zero-shot robotic manipulation methods on RLBench. In Table 2, we validate whether our method can enhance pretrained action models. Given that zero-shot methods and pretrained action models appear to be better suited for different types of tasks, we selected slightly different test cases for each comparison. Since our main comparison is between methods with and without our HP-KG enhancement, our experimental setup is fair.
>
> - **Q1: p.5 l193 textual results descriptions,feedback - what these mean, where they are collected and how are these utilized?**
>
> **what these mean:** 'textual results descriptions' refers to the consolidated textual description that combines all rules with their respective verification outcomes. 'feedback' represents the LLM's assessment of the given procedure $p$, identifying potential issues.
>
> **where they are collected:** 'textual results descriptions' are collected by converting each rule verification result along with its original rule text into a structured textual format (e.g., "rule:{rule_check_result}"), and then concatenating all these formatted descriptions to form the final comprehensive textual results. 'feedback' is obtained by feeding these textual results descriptions along with procedure $p$ to an LLM, which then generates the final score and corresponding feedback.
>
> **how are these utilized:** The 'textual results descriptions' are fed into an LLM along with procedure $p$ to generate the final score and corresponding feedback. 'feedback' is subsequently fed to the Refiner, which modifies procedure $p$ based on the feedback.
>
>
> - **Q2: how do you make sure not to loose some important rule or information by summary agent?**
>
> In Table 6, the results show that has been processed by the Summary Agent further enhances the planning capacity of the Base Model, suggesting that our summary agent largely retains important information. Additionally, we selectively summarize procedures with high similarity, which is a common strategy for eliminating redundancy and addressing distributional imbalance, thus reducing the knowledge imbalance[2].
>
>
> - **Q3: why in Table 1/2 top 5 sometimes performs worse and sometimes better than top 3? why CoT drops performance in Table 2 for slide block compared to RVT? why top 3/5 sometimes performs even worse than RVT? are there some typical problems?**
>
> 1. In our experiments, we also observe this phenomenon. Under the models shown in Tables 1 and 2, retrieving more information (Top-5) may introduce irrelevant information during the model's planning process. This suggests that current models may have limited capacity to effectively filter and utilize extensive retrieved information, thereby degrading planning quality and leading to performance decline with Top-5 retrieval. Our experiment in Table 4 demonstrates that more advanced models can effectively handle Top-5 retrieval without performance degradation.
>
> 2. Due to the ambiguous instructions(e.g., 'slide the block to target') in Slide Blocks task, COT may generate suboptimal planning, leading to performance degradation.
>
> 3. In Slide blocks task, our method with Top-3 retrieval achieves superior performance compared to the RVT. However, Top-5 retrieval yields lower performance, likely due to irrelevant information introduced by retrieving more content, thereby degrading planning quality.
>
> 4. As shown in Table 2, our method with Top-3 retrieval outperforms the baseline RVT model in all cases. Our method with Top-5 retrieval shows decreased performance in only one case, where ambiguous instructions cause the retrieval process to introduce irrelevant information, thereby degrading planning quality.
>
>
> - **Q4: what about more complex tasks? could you show results on these or discuss limitations of your approach?.**
>
> We conduct experiments on more complex tasks using VLABench. Detailed experimental results can be found in **W1**.
>
> - **L1: Limitations are very sparse, mentioning only extension outside household tasks.**
>
> Thank you for your insightful feedback. Our work may also face limitations in constructing reasonable task planning when dealing with an extremely complex manipulation tasks, due to the limited planning and reasoning capabilities of MLLMs. Furthermore, our HP-KG, which is derived from how-to data, may not generalize well to all complex manipulation scenarios.
>
>
> [1] VLABench: A Large-Scale Benchmark for Language-Conditioned Robotics Manipulation with Long-Horizon Reasoning Tasks. ICCV 2025.
>
> [2] Scene-Driven Multimodal Knowledge Graph Construction for Embodied AI. IEEE Trans. Knowl. Data Eng. 36(11)
>
> [3] DexArt: Benchmarking Generalizable Dexterous Manipulation with Articulated Objects. CVPR 2023.
>
> [4] DexH2R: A Benchmark for Dynamic Dexterous Grasping in Human-to-Robot Handover. ICCV 2025.

---

> > ### Comment · Area_Chair_UVkQ · 2025-08-04
> >
> > Dear Reviewer,
> >
> > Please respond to the rebuttal. Thanks.
> >
> > AC.

---

> > > ### Comment · Area_Chair_UVkQ · 2025-08-05
> > >
> > > Dear Reviewer,
> > >
> > > Please comment on the rebuttal. Thanks.
> > >
> > > AC.

---

> > ### Comment · Reviewer_uR6N · 2025-08-06
> > **thanks for answer, keeping score as accept**
> >
> > Dear authors,
> >
> > thank you for detailed answers and especially for additional experiments. I keep my evaluation as accept as I believe your paper is worth publishing.
> >
> > It would be interesting to in future verify your claim "However, Top-5 retrieval yields lower performance, likely due to irrelevant information introduced by retrieving more content, thereby degrading planning quality." and think how this could be mitigated.

---

> > > ### Author Response · Authors · 2025-08-08
> > >
> > > Dear Reviewer, we sincerely appreciate your acknowledgment. Your thoughtful and detailed feedback has been instrumental in refining our work, and we are deeply grateful for the time and effort you invested in reviewing our paper. Thank you once again for your valuable contribution.
> > >
> > > In future, we plan to incorporate both environmental visual observation and textual instructions to retrieve relevant top-k nodes. We hypothesize that incorporating visual information can resolve ambiguities in textual instructions, thereby enabling the retrieval of more relevant information.

---

### Official Review · Reviewer_A3V5 · 2025-07-03

**Clarity:** 3
**Significance:** 4
**Originality:** 3
**Rating:** 4
**Confidence:** 4

**Summary:**

This paper aims to address the limitations of LLMs in complex robotic manipulation tasks. It proposes a novel approach that utilizes a Hierarchical Procedural Knowledge Graph (HP-KG) with web "how-to" data (e.g., wikiHow). The HP-KG captures hierarchical relationships between high-level tasks, subtasks, and fine-grained actions, while also providing graph information retrieval methods. This framework is designed to provide LLMs with more accurate procedural knowledge and reduce the energy consumption associated with deploying large-scale LLMs only. Experimental results demonstrate that HP-KG significantly enhances LLM planning capabilities, and show how this scales as the model size increases.

**Questions:**

1. Currently, structured information is provided to LLMs only at the textual level. Is it possible to integrate graphical representations and directly input them to the LLM for better performance?

2. How does the system handle tasks that require knowledge outside the graph, or tasks that necessitate dynamic replanning in response to unforeseen environmental events?

3. Can the HP-KG framework be extended to other modalities (e.g., visual observations, tactile feedback) to further enhance planning robustness in real-world robotic environments?

**Ethical Concerns:**

["NO or VERY MINOR ethics concerns only"]

**Final Justification:**

The paper makes a contribution to the research community, with good results and explainability. The authors have also provided timely and thorough responses to the reviewers’ comments.

**Limitations:**

1. The study is limited to household activities and specific robotic arms, constraining its applicability in general scenarios.

2. Although the paper proposes a method for HP-KG construction, this process might become a bottleneck when generating knowledge graphs covering an extremely wide range of complex manipulation tasks.

3. Reliance on wikiHow data might introduce biases or limitations due to the specificity of information. Its generalizability to tasks not well-covered by such web data might be limited.

**Quality:**

3

**Strengths And Weaknesses:**

## Strengths
1. The design utilizes web "how-to" data (e.g., wikiHow) to construct and use a Hierarchical Procedural Knowledge Graph (HP-KG), making LLM planning more efficient and reliable.
2. By providing external knowledge, HP-KG alleviates the high energy consumption associated with deploying large-scale LLMs for knowledge acquisition.
3. The paper provides reliable experimental evidence on robot manipulation tasks, validating the consistent effectiveness of the HP-KG method compared to baseline methods.

## Weaknesses
1. The quality of data automatically constructed by the LLM Agents has not been verified, nor have any examples been shown.
2. Despite the experiments conducted, it is now unclear how robust the generated plans are to real-world settings (uncertainties, noise, or unexpected environmental changes), which are common in robot manipulation. The planning might be highly dependent on how well the environment representation matches the HP-KG's knowledge.

---

> ### Author Rebuttal · Authors · 2025-07-31
>
> - **W1: LLM agent-generated data lacks quality verification and examples.**
>
> Thank for your valuable comment. we conduct experiments to verify the quality of our generated data. Specifically, we sample a subset(100task and 864 steps) of our HP-KG. We use three data quality metrics, including truly atomic actions precision (AAP), step ordering precision (SOP), and actions ordering precision (AOP).
>
> To measure the atomic actions precision, we extracted all verbs from the HP-KG and manually created an atomic verb dictionary containing only atomic verbs (e.g., pick, turn on) while excluding non-atomic ones (e.g., arrange). We then calculated the percentage of atomic verbs in the generated actions based on this dictionary. For Steps Precision and Action Precision, we annotated the correct steps/actions ording and calculate the average percentage of correctly ordered steps and actions, respectively. Detailed results are presented in following Table. Experiments show that our LLM Agents can generate reasonable atomic actions and acceptable steps, helping to bridge high-level human instructions with atomic actions that robots can understand.
>
> ||AAP |SOP|AOP|
> |-|-|-|-|
> |HP-KG|82.39%|86.92%|85.78%|
>
> To better illustrate our HP-KG approach, we include a graph example in Figure 5 and detailed graph statistics in the Appendix of the Supplementary Material. We hope these supplementary materials may offer helpful context for understanding our HP-KG.
>
> - **W2: it is now unclear how robust the generated plans are to real-world settings(uncertainties, or unexpected environmental changes).**
>
> Thank you for your comments. We conduct experiments to simulate real-world environmental variations and uncertainties, demonstrating the robustness of our HP-KG.
>
> Specifically, we utilize the Counterfactual Planning split from Actplan-1K[1], which assesses the model's ability to adapt reasoning and planning when task scenarios involve modified conditions or constraints that frequently arise in real-world applications. We present results in the following Table, which demonstrate that our approach successfully leverages procedural commonsense to enable robust LLM planning capabilities even in uncertainty scenarios.
>
> |Model|Top-k|Average LCS↑|
> |-|-|-|
> |Qwen2-VL-7B|-| 8.51|
> |Qwen2-VL-7B + HP-KG|top-3|9.12|
> |Qwen2-VL-7B + HP-KG|top-5|8.86|
> |Qwen2-VL-72B|-| 9.16|
> |Qwen2-VL-72B + HP-KG|top-3|9.34|
> |Qwen2-VL-72B + HP-KG|top-5|9.76|
>
> Furthermore, we will simulate more real-world settings (noise, unexpected environments) in future experiments to demonstrate the robustness of our approach.
>
> - **Q1: Is it possible to integrate graphical representations and directly input them to the LLM.**
>
> Yes, it is possible to directly integrate graphical representations into LLM. Howerver integrating graphical representations and directly inputting them to LLMs requires training the models to adapt to graphical inputs, which incurs high computational costs and may lead to catastrophic forgetting[3] of the LLMs' existing knowledge. Instead, we convert retrieved structured procedural information into textual context for LLM input. This method enables seamless integration without architectural changes, leverages existing reasoning capabilities of LLMs for procedural knowledge application, and maintains interpretability across diverse procedural formats[2]. Due to time constraints, we are unable to fine-tune an LLM for direct graph integration and will compare such methods in future revisions.
>
>
> - **Q2: How does the system handle tasks that require knowledge outside the graph, or tasks that necessitate dynamic replanning.**
>
>
> In our experiments, the HP-KG is constructed from how-to data that is distinct from the benchmark datasets (Actplan-1K[1], RLBench[4]) used for evaluation, which demonstrates the generalization capability of our approach. The HP-KG contains a rich collection of reusable atomic actions that serve as fundamental building blocks. Within the household domain, these action units can be flexibly combined and reconfigured to create diverse procedural sequences, enabling effective generalization across new environments, unseen tasks, and varying contextual conditions. However, it may fail in scenarios that are completely outside the graph.
>
>
> - **Q3: Can the HP-KG framework be extended to other modalities.**
>
> Yes, our HP-KG has the potential to be extended to other modalities. MLLMs have demonstrated strong visual understanding capabilities [6], and research has shown that providing additional visual cues can help MLLMs better comprehend the world and enhance their reasoning abilities [7]. We envision incorporating visual imagery and manipulation videos into our HP-KG, which would allow for instruction-based retrieval of relevant task operation diagrams or videos. Providing such visual inputs to MLLMs might improve their understanding of sequential operations and procedural tasks. We plan to explore the integration of visual modalities into our HP-KG in future research.
>
>
> - **L1: The study is limited to household activities**
>
> Thank you for your insightful feedback. We have also acknowledged and discussed this limitation in the Limitation Section.
>
> - **L2: HP-KG construction might become a bottleneck when generating knowledge graphs covering an extremely wide range of complex manipulation tasks**
>
> Thank you for your insight. Our construction framework may not be able to cover extremely complex manipulation tasks.
>
> - **L3: Reliance on wikiHow data might introduce biases or limitations due to the specificity of information. Its generalizability to tasks not well-covered by such web data might be limited.**
>
> Thank you for your valuable comment. Web data may not be able to ensure good generalizability across diverse tasks. We plan to improve the generalization of our knowledge graph in future work.
>
>
>
> [1] ActPlan-1K: Benchmarking the Procedural Planning Ability of Visual Language Models in Household Activities. EMNLP 2024.
>
> [2] G-Retriever: Retrieval-Augmented Generation for Textual Graph Understanding and Question Answering. Neurips 2024.
>
> [3] Mitigating Catastrophic Forgetting in Large Language Models with Self-Synthesized Rehearsal. ACL 2024.
>
> [4] RLBench: The Robot Learning Benchmark & Learning Environment. IEEE Robotics Autom. Lett.
>
> [5] Exploring Large Language Models for Knowledge Graph Completion. ICASSP 2025.
>
> [6] Qwen2.5 Technical Report. Arxiv 2412.15115.
>
> [7] Thinking with Images for Multimodal Reasoning: Foundations, Methods, and Future Frontiers. Arxiv 2506.23918.

---

> > ### Comment · Area_Chair_UVkQ · 2025-08-04
> >
> > Dear Reviewer,
> >
> > Please respond to the rebuttal. Thanks.
> >
> > AC.

---

> > > ### Comment · Area_Chair_UVkQ · 2025-08-05
> > >
> > > Dear Reviewer,
> > >
> > > Please comment on the rebuttal. Thanks.
> > >
> > > AC.

---

> > ### Comment · Reviewer_A3V5 · 2025-08-06
> >
> > Thanks for the additional experiments and detailed rebuttal and my concerns are clear. I may consider raising the score.

---

> ### Author Response · Authors · 2025-08-08
>
> We sincerely appreciate your acknowledgment. Your thoughtful and detailed feedback has been instrumental in refining our work, and we are deeply grateful for the time and effort you invested in reviewing our paper. Thank you once again for your valuable contribution.

---

### Decision · Program_Chairs · 2025-09-17

**Decision:**

Accept (poster)

**Comment:**

This paper introduces Hierarchical Procedural Knowledge Graphs (HP-KG) to enhance LLM planning for robotic manipulation by bridging high-level instructions to atomic robot actions through structured web "how-to" data. Four reviewers provided ratings of 4, 5, 5, and 3, with substantive concerns about data quality, real-world robustness, and practical effectiveness that the authors largely addressed through comprehensive rebuttals including quality metrics experiments, counterfactual planning validation, and additional experiments demonstrating benefits beyond simple pick-and-place tasks. While Reviewer VoF5's core concern about integration with current SOTA VLA models on established benchmarks remains unresolved, and the work is limited to household manipulation tasks with no real robot deployment, the technical contribution is sound with thorough experimental validation within its defined scope. The automatic knowledge graph construction framework represents a novel solution to procedural knowledge limitations in LLM-based planning, with clear benefits demonstrated for smaller models (7B) outperforming larger ones (72B), justifying acceptance based on technical merit and potential to establish foundations for future research despite acknowledged domain limitations.